# Drop-DTW: Aligning Common Signal Between Sequences While Dropping Outliers

**Nikita Dvornik [1,2]  Isma Hadji [1]  Konstantinos G. Derpanis [1]  Animesh Garg [2]  Allan D. Jepson [1]**

[1]Samsung AI Centre Toronto
[2]University of Toronto, Vector Institute
{isma.hadji, allan.jepson}@samsung.com
{n.dvornik, k.derpanis}@partner.samsung.com
garg@cs.toronto.edu

## Abstract

In this work, we consider the problem of sequence-to-sequence alignment for signals containing outliers. Assuming the absence of outliers, the standard Dynamic Time Warping (DTW) algorithm efficiently computes the optimal alignment between two (generally) variable-length sequences. While DTW is robust to temporal shifts and dilations of the signal, it fails to align sequences in a meaningful way in the presence of outliers that can be arbitrarily interspersed in the sequences. To address this problem, we introduce Drop-DTW, a novel algorithm that aligns the common signal between the sequences while automatically dropping the outlier elements from the matching. The entire procedure is implemented as a single dynamic program that is efficient and fully differentiable. In our experiments, we show that Drop-DTW is a robust similarity measure for sequence retrieval and demonstrate its effectiveness as a training loss on diverse applications. With Drop-DTW, we address temporal step localization on instructional videos, representation learning from noisy videos, and cross-modal representation learning for audio-visual retrieval and localization. In all applications, we take a weakly- or unsupervised approach and demonstrate state-of-the-art results under these settings.

## 1 Introduction

The problem of sequence-to-sequence alignment is central to many computational applications. Aligning two sequences (e.g., temporal signals) entails computing the optimal pairwise correspondence between the sequence elements while preserving their match orderings. For example, video [1] or audio [2] synchronization are important applications of sequence alignment in the same modality, while the alignment of video to audio [3] represents a cross-modal task. Dynamic Time Warping (DTW) [2] is a standard algorithm for recovering the optimal alignment between two variable length sequences. It efficiently solves the alignment problem by finding all correspondences between the two sequences, while being robust to temporal variations in the execution rate and shifts.

A major issue with DTW is that it enforces correspondences between *all* elements of both sequences and thus cannot properly handle sequences containing outliers. That is, given sequences with interspersed outliers, DTW enforces matches between the outliers and clean signal, which is prohibitive in many applications. A real-world example of sequences containing outliers is instructional videos. These are long, untrimmed videos depicting a person performing a given activity (e.g., making a latte) following a pre-defined set of ordered steps (e.g., a recipe). Typically, only a few frames in the video correspond to the instruction steps, while the rest of the video frames are unrelated to the main activity (e.g., the person talking); see Figure 1 for an illustration. In this case, matching the outlier frames to instruction steps will not yield a meaningful alignment. Moreover, the match score

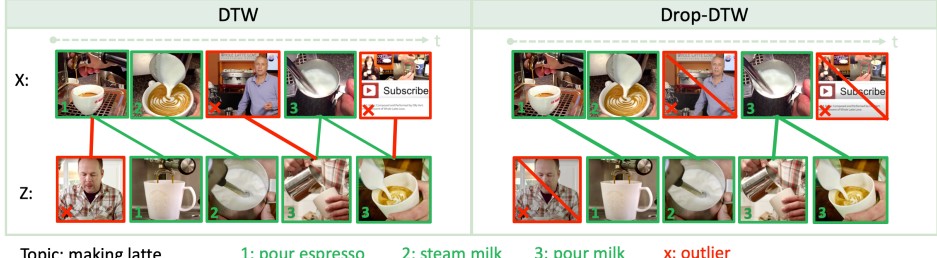

**Figure 1: Aligning instructional videos.** Left: Both video sequences (top and bottom) depict the three main steps of "making latte"; however, there are unrelated video segments, i.e., outliers, inbetween the steps. DTW aligns all the frames with each other and creates false correspondences where outliers are matched to signal (red links). Right: In contrast, Drop-DTW finds the optimal alignment, while *simultaneously* dropping unrelated frames (crossed out), leaving only correct correspondences (green links).

between such sequences, computed by DTW, will be negatively impacted by "false" correspondences and therefore cannot be used reliably for downstream tasks, e.g., retrieval or representation learning.

In this paper, we introduce Drop-DTW to address the problem of matching sequences that contain interspersed outliers as illustrated in Figure 1 (right) and compared to standard DTW (on the left). While various improvements to DTW have been previously proposed (e.g., [1, 4–8]), Drop-DTW is the first to augment DTW with the ability to flexibly skip through irrelevant parts of the signal *during* alignment, while still allowing one-to-many matching. Rather than relying on a two-step greedy approach, where elements are first dropped before aligning the remaining signal, Drop-DTW achieves this in a unified framework that solves for the optimal temporal alignment while *jointly* detecting outliers. Drop-DTW casts sequence alignment as an optimization problem with a novel component specifying the cost of dropping an element within the optimization process. It is efficiently realized using a dynamic program that naturally admits a differentiable approximation and can be efficiently used at training and inference time.

**Contributions.**

- We propose an extension of DTW that is able to identify and align the common signal between sequences, while *simultaneously* excluding interspersed outliers from the alignment.

- The proposed Drop-DTW formulation naturally admits a differentiable approximation and we therefore demonstrate its usage as a training loss function.

- We demonstrate the utility of Drop-DTW, for both training and inference for multi-step localization in instructional videos, using only ordered steps as weak supervision. We achieve state-of-the-art results on the CrossTask [9] dataset and are the first to tackle the COIN [10] and YouCook2 [11] datasets, given only ordered steps (i.e., no framewise labels are used).

- We employ Drop-DTW as a loss function for weakly-supervised video representation learning on the PennAction dataset [12], modified to have interspersed outliers, and unsupervised audio-visual representation learning on the AVE dataset [13]. Compared to the baselines, Drop-DTW yields superior representations, as measured by its performance on various downstream tasks.

Our code is available at: https://github.com/SamsungLabs/Drop-DTW.

## 2  Related work

**Sequence alignment.** The use of aligned sequences in learning has recently seen a growing interest across various tasks [14, 4, 5, 15, 1, 6, 16, 17]. While some methods start from aligned paired signals [14, 18], others directly learn to align unsynchronized signals [15, 5, 1, 6]. One approach tackles the alignment problem locally by maximizing the number of one-to-one correspondences using a soft nearest neighbors method [15]. More closely related are methods [4, 5, 1, 16] that seek a global alignment between sequences by relying on Dynamic Time Warping (DTW) [2]. To handle noise in the feature space some methods use Canonical Correlation Analysis (CCA) with standard DTW [19, 20]. To use DTW for end-to-end learning, differentiable approximations of the discrete operations (i.e., the min operator) in DTW have been explored [4, 1]. One of the main downsides of standard DTW-based approaches is that they require clean, tightly cropped, data with matching

endpoints. In contrast to such methods, Drop-DTW extends DTW and its differentiable variants with the ability to handle outliers in the sequences by allowing the alignment process to automatically skip outliers (i.e., match signal while dropping outliers). While previous work [21, 6, 22] targeted rejecting outliers limited to the start or end of the sequences, our proposed Drop-DTW is designed to handle outliers interspersed in the sequence, including at the endpoints. Other work proposed limited approaches to handle such outliers either using a greedy two-step approach [7], consisting of outlier rejection followed by alignment, or by restricting the space of possible alignments [8, 23] (e.g., only allowing one-to-one matches and individual drops). In contrast, Drop-DTW solves for the optimal temporal alignment while simultaneously detecting outliers and allowing for one-to-many matches. As shown in our experiments, the ability of Drop-DTW to flexibly skip outliers during alignment can in turn be used at inference time to localize the start and end times of inlier subsequences.

**Representation learning.** Self- and weakly-supervised approaches are increasingly popular for representation learning as they eschew the need of dense labels for training. Such approaches are even more appealing for video-based tasks that typically require inordinately large labeling efforts. A variety of (label-free) proxy tasks have emerged for the goal of video representation learning [24–41].

More closely related are methods using sequence alignment as a proxy task [5, 15, 1, 6, 16, 17]. Our approach has the added advantage of handling interspersed outliers during alignment. Also related are methods that leverage multimodal aspects of video, e.g., video-audio [42, 43] or video-language alignment [44–47]. Our step localization application builds on strong representations learned via the task of aligning vision and language using narrated instructional videos [45]. However, we augment these representations with a more global and robust approach to the alignment process, thereby directly enabling fine-grained applications, such as multi-step localization.

**Step localization.** Temporal step localization consists of determining the start and end times of one or more activities present in a video. These methods can be broadly categorized into two classes based on the level of training supervision involved. Fully supervised approaches (e.g., [48, 49, 10]) rely on fine-grained temporal labels indicating the start and end times of activities. Weakly supervised approaches eschew the need for framewise labels. Instead, they use a video-level label corresponding to the activity category [50, 51], or sparse time stamps that provide the approximate (temporal) locations of the activities. More closely related are methods that rely on the order of instruction steps in a clip to yield framewise step localization [52–54, 5, 9]. In contrast, we do not rely on categorical step labels from a fixed set of categories and demonstrate that our approach applies to any ordered sequence of embedded steps, such as embedded descriptions in natural language.

## 3 Technical approach

### 3.1 Preliminaries: Dynamic time warping

Dynamic Time Warping (DTW) [2] computes the optimal alignment between two sequences subject to certain constraints. Let $X = [x_1, \ldots, x_N]^\top \in \mathbb{R}^{N \times d}$ and $Z = [z_1, \ldots, z_K]^\top \in \mathbb{R}^{K \times d}$ be the input sequences, where $N, K$ are the respective sequence lengths and $d$ is the dimensionality of each element. The valid alignments between sequences are defined as a binary matrix, $M \in \{0, 1\}^{K \times N}$, of the pairwise correspondences, where $M_{i,j} = 1$ if $z_i$ is matched to $x_j$, and $0$ otherwise. Note that $Z$ corresponds to the first (row) index of $M$ and $X$ the second (column). Matching an element $z_i$ to element $x_j$ has a cost $C_{i,j}$ which is typically a measure of dissimilarity between the elements. DTW yields the optimal alignment $M^*$ between sequences $Z$ and $X$ that minimizes the overall matching cost:

$$M^* = \underset{M \in \mathcal{M}}{\arg\min} \langle M, C \rangle = \underset{M \in \mathcal{M}}{\arg\min} \sum_{i,j} M_{i,j} C_{i,j}, \tag{1}$$

where $\langle M, C \rangle$ is the Frobenius inner product and $\mathcal{M}$ is the set of all feasible alignments that satisfy the following constraints: monotonicity, continuity, and matching endpoints ($M_{1,1} = M_{K,N} = 1, \forall M \in \mathcal{M}$). Fig. 2 (a) provides an illustration of feasible and optimal alignments. The cost of aligning two sequences with DTW is defined as the cost of the optimal matching: $c^* = \langle M^*, C \rangle$. DTW proposes an efficient dynamic programming algorithm to find the solution to (1).

In the case of sequences containing outliers, the matching endpoints and continuity constraints are too restrictive leading to unmeaningful alignments. Thus, we introduce Drop-DTW that admits more flexibility in the matching process.

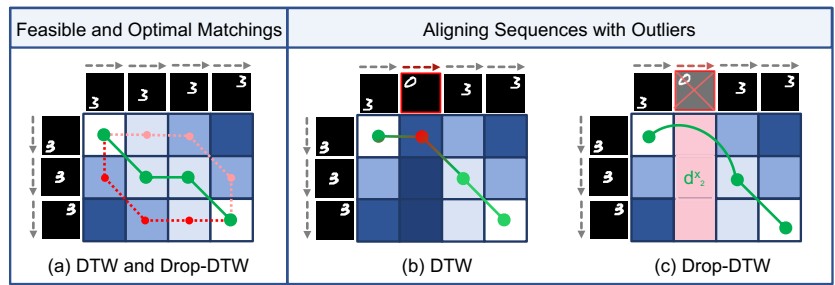

**Figure 2: Optimal alignment with DTW and Drop-DTW**. Aligning two different videos where the digit "3" moves across the square frame. The colored matrices represent the pairwise matching costs, $C$, with darker cells indicating higher cost, $C_{i,j}$. The paths on the grid are alignment paths, while the points on them indicate a pairwise match between the corresponding row and column elements. **(a)** All three paths are feasible DTW paths, while only one of them (with solid line in green) is optimal. **(b)** When sequence $X$ contains an outlier (i.e., digit "0"), DTW uses it in the alignment and incurs a high cost (red point). **(c)** In contrast, Drop-DTW skips the outlier (while paying the cost $d_2^x$) and only keeps the relevant matches.

### 3.2 Sequence alignment with Drop-DTW

Drop-DTW extends consideration of feasible alignments, $\mathcal{M}$, to those adhering to the monotonicity constraint only. Consequently, unlike DTW, elements can be dropped from the alignment process.

Before introducing Drop-DTW, let us discuss a naive solution to the problem of outlier filtering in DTW. One could imagine a greedy two-step approach where one first (i) drops the outliers, and then (ii) aligns remaining elements with standard DTW. Since step (i) (i.e., dropping) is performed independently from step (ii) (i.e., alignment), this approach yields a sub-optimal solution. Critically, if an important element is erroneously dropped in step (i), it is impossible to recover it in step (ii). Moreover, the outlier rejection step is order agnostic and results in drops broadly scattered over the entire sequence, which makes this approach inapplicable for precise step localization. To avoid such issues, it is critical to jointly address (i) outlier detection and (ii) sequence alignment.

For this purpose, we propose Drop-DTW, a unified framework that solves for the optimal temporal alignment while jointly enabling element dropping by adding another dimension to the dynamic programming table. Specifically, dropping an element $x_j$ means there is no element in $Z$ that has been matched to $x_j$. This is captured in the $j$-th column of the correspondence matrix $M$ containing only zeros, i.e., $M_{:,j} = \mathbf{0}$.

To account for unmatched elements in the alignment objective, we extend the set of costs beyond pairwise matching, $C_{i,j}$, used in DTW, with novel drop costs $d_i^z \in \mathbb{R}^K$ and $d_j^x \in \mathbb{R}^N$, for elements $z_i \in Z$ and $x_j \in X$, respectively. The optimal matching can be then defined as follows:

$$M^* = \underset{M \in \bar{\mathcal{M}}}{\arg \min} \langle M, C \rangle + P_z(M) \cdot (d_i^z) + P_x(M) \cdot (d_j^x), \tag{2}$$

where $\cdot$ denotes the inner product, $\bar{\mathcal{M}}$ is the set of binary matrices satisfying just the monotonicity constraint, and $P_x(M)$ is a vector with the $j$-th element equal to one if the $M_{:,j} = \mathbf{0}$ and zero otherwise; $P_z(M)$ is defined similarly, but on rows.

For clarity, in Algorithm 1, we describe the dynamic program for efficiently computing the optimal alignment and its cost with dropping elements limited to $X$, i.e., $d_i^z = \infty$. Our general Drop-DTW algorithm that drops elements from *both* sequences, $X$ and $Z$, is given in the supplemental.

### 3.3 Definition of match costs

The pairwise match costs, $C_{i,j}$, are typically defined based on the dissimilarity between elements $x_j$ and $z_i$. In this paper, we consider two different ways to define the pairwise costs, depending on the application. In general, when dropping elements from both $X$ and $Z$ is permitted, we consider the following symmetric match cost:

$$C_{i,j}^s = 1 - \cos(z_i, x_j). \tag{3}$$

**Algorithm 1** Subsequence alignment with Drop-DTW.

---

1: **Inputs**: $C \in \mathbb{R}^{K \times N}$ - pairwise match cost matrix, $d^x$ - drop costs for elements in $x$.
2:                                                       ▷ *initializing dynamic programming tables*
3: $D^+_{0,0} = 0; D^+_{i,0} = \infty; D^+_{0,j} = \infty;$        $i \in [\![K]\!], j \in [\![N]\!]$                 ▷ *match table*
4: $D^-_{0,0} = 0; D^-_{i,0} = \infty; D^-_{0,j} = \sum_{k=1}^{j} d^x_k;$    $i \in [\![K]\!], j \in [\![N]\!]$                  ▷ *drop table*
5: $D_{0,0} = 0; D_{i,0} = D^-_{i,0}; D_{0,j} = D^-_{0,j};$      $i \in [\![K]\!], j \in [\![N]\!]$             ▷ *optimal solution table*
6: **for** $i = 1, \ldots, K$ **do**                                    ▷ *iterating over elements in $Z$*
7:      **for** $j = 1, \ldots, N$ **do**                              ▷ *iterating over elements in $X$*
8:          $D^+_{i,j} = C_{i,j} + \min\{D_{i-1,j-1}, D_{i,j-1}, D^+_{i-1,j}\}$          ▷ *consider matching $z_i$ to $x_j$*
9:          $D^-_{i,j} = d^x_j + D_{i,j-1}$                           ▷ *consider dropping $x_j$*
10:          $D_{i,j} = \min\{D^+_{i,j}, D^-_{i,j}\}$                     ▷ *select the optimal action*
11:      **end for**
12: **end for**
13: $M^* = \text{traceback}(D)$        ▷ *compute the optimal alignment by tracing back the minimum cost path*
14: **Output:** $D_{K,N}, M^*$

---

Alternatively, when one of the sequences, i.e., $Z$, is known to contain only signal, we follow [1], who show that the following asymmetric matching cost is useful during representation learning:

$$C^a_{i,j} = -\log(\text{softmax}_1(Z^\top X/\gamma))_{i,j}, \tag{4}$$

where $\text{softmax}_1(\cdot)$ defines a standard softmax operator applied over the first tensor dimension.

Importantly, the matching cost $C_{i,j}$ does not solely dictate whether the elements $x_j$ and $z_i$ are matched. Instead, the optimal matching is governed by (2) where the match cost $C_{i,j}$ is just one of the costs affecting the optimal alignment, along with drop costs $d^x_j$ and $d^z_i$.

### 3.4 Definition of drop costs

There are many ways to define the drop costs. As a starting point, consider a drop cost that is a fixed constant across all elements in $X$ and $Z$: $d^x_j = d^z_i = s$. Setting the constant $s$ too low (relative to the match costs, $C$) will lead to a high frequency of drops and thus matching only a small signal fraction. In contrast, setting $s$ too high may result in retaining the outliers, i.e., no drops.

**Percentile drop costs.** To avoid such extreme outcomes, we define the drop cost on a per-instance basis with values comparable to those of the match cost, $C$. In particular, we define the drop costs as the $p \in [0, 100]$ top percentile of the values contained in the cost matrix, $C$:

$$s = \text{percentile}(\{C_{i,j} \mid 1 \leq i \leq K, 1 \leq j \leq N\}, p). \tag{5}$$

Defining the drop cost as a function of the top percentile match costs has the advantage that adjusting $p$ allows one to inject a prior belief on the outlier rate in the input sequences.

**Learnable drop costs.** While injecting prior knowledge in the system using the percentile drop cost can be advantageous, it may be hard to do so when the expected level of noise is unknown or changes from one dataset to another. To address this scenario, we introduce an instantiation of a leanable drop cost here. We choose to define the outliers drop costs in $X$ based on the content of sequence $Z$. This is realized as follows:

$$d^x = X \cdot f_{\omega_x}(\bar{Z}); \quad d^z = Z \cdot f_{\omega_z}(\bar{X}), \tag{6}$$

$\bar{Z}, \bar{X}$ are the respective means of sequences $Z, X$, and $f_\omega(\cdot)$ is a learnable function (i.e., a feed-forward neural net) parametrized by $\omega$. This definition can yield a more adaptable system.

### 3.5 Drop-DTW as a differentiable loss function

**Differentiable approximation.** To make the matching process differentiable, we replace the hard min operator in Alg. 1 with the following differentiable approximation introduced in [1]:

$$\text{smoothMin}(\boldsymbol{x}; \gamma) = \boldsymbol{x} \cdot \text{softmax}(-\boldsymbol{x}/\gamma) = \frac{\boldsymbol{x} \cdot e^{-\boldsymbol{x}/\gamma}}{\sum_j e^{-x_j/\gamma}}, \tag{7}$$

where $\gamma > 0$ is a hyperparameter controlling trade-off between smoothness and approximation error.

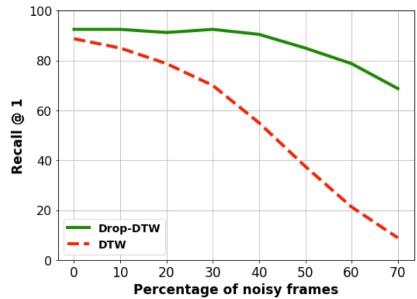

**Figure 3: Drop-DTW for retrieval on TMNIST.** We consider queries, $Z$, from TMNIST-part with interspersed noise, for indexing into TMNIST-full. We show that Drop-DTW is more robust to interspersed noise.

**Loss function.** The differentiable Drop-DTW is a sequence match cost suitable for representation learning (cf. [1]). The corresponding Drop-DTW loss function is defined as follows:

$$\mathcal{L}_{\text{DTW}}(Z, X) = \text{Drop-DTW}(Z, X) = D_{K,N}(Z, X), \tag{8}$$

where $D_{K,N}(Z, X)$ is the optimal match cost between $Z$ and $X$ computed in Alg. 1.

## 4 Experiments

To demonstrate the advantages of Drop-DTW, we first present a controlled experiment using synthetic data to demonstrate the robustness of Drop-DTW (Sec. 4.1). Next, we show the strength of Drop-DTW on various applications, including multi-step localization (Sec 4.2), representation learning from noisy videos (Sec 4.3), and audio-visual alignment for retrieval and localization (Sec 4.4).

### 4.1 Controlled synthetic experiments

**Synthetic dataset.** We use the MNIST dataset [55] to generate videos of moving digits (cf. [56]). Each video in our dataset depicts a single digit moving along a given trajectory. For each digit-trajectory pair, we generate two videos: (i) the digit moves along the full trajectory (termed TMNIST-full) and (ii) the digit performs a random sub-part of the trajectory (termed TMNIST-part). Each video frame is independently encoded with a shallow ConvNet [55] trained for digit recognition. We elaborate on the the dataset and the framewise embedding network in the supplemental. We use these datasets for sequence retrieval. We also show results on subsequence localization in the supplemental.

**Video retrieval.** In this experiment, we use TMNIST-part as queries and look for the most similar videos in TMNIST-full. A correct retrieval is defined as identifying the video in TMNIST-full containing the same digit-trajectory pair as the one in the query. We use the Recall@1 metric to report performance. To analyze the strength of Drop-DTW in this controlled setting, we also introduce temporal noise in the query sequence by randomly blurring a subset of the video frames. We compare Drop-DTW to standard DTW [2] in this experiment. In all cases, we use the alignment score obtained from each algorithm as a matching cost between sequences. For this experiment, we use the symmetric matching costs defined in (3). Since no training is involved in this experiment, we set the drop costs to a constant, $d^x = d^z = 0.3$, which we establish through cross-validation.

Figure 3 compares the performance of the two alignment algorithms with various levels of noise. As expected, DTW is very sensitive to interspersed noise. In contrast, Drop-DTW remains robust across the range of noise levels. Concretely, for the highest noise level in Fig. 3, we see that Drop-DTW is $8\times$ better than DTW, which speaks decisively in favor of Drop-DTW.

### 4.2 Multi-step localization

We now evaluate Drop-DTW on multi-step localization in more realistic settings. For this task, we are given as input: (i) a long untrimmed video of a person performing a task (e.g., "making salad"), where the steps involved in performing the main task are interspersed with irrelevant actions (e.g., intermittently washing dishes), and (ii) an ordered set of textual descriptions of the main steps (e.g., "cut tomatoes") in this video. The goal is to temporally localize each step in the video.

**Datasets.** For evaluation, we use the following three recent instructional video datasets: CrossTask [9], COIN [10], and YouCook2 [11]. Both CrossTask and COIN include instructional videos of different activity types (i.e., tasks), with COIN being twice the size of CrossTask in the number of videos and spanning 10 times more tasks. YouCook2 is the smallest of these datasets and focuses on cooking tasks. While all datasets provide framewise labels for the start and end times of each step in a video, we take a weakly-supervised approach and only use the ordered step information.

**Metrics.** We evaluate the performance of Drop-DTW on step localization using three increasingly strict metrics. **Recall** [9] is defined as the number of steps correctly assigned to the ground truth time interval divided by the total number of steps and is the least strict metric out of the three considered. **Framewise accuracy (Acc.)** [10] is defined as the ratio between the number of frames assigned the correct step label (including background) and the total number of frames. Finally, **Intersection over Union (IoU)** [11] is defined as the sum of the intersections between the predicted and ground truth time intervals of each step divided by the sum of their unions. IoU is the most challenging of the three metrics as it more strictly penalizes misalignments.

**Training and Inference.** We start from (pretrained) strong vision and language embeddings obtained from training on a large instructional video dataset [45]. We further train a two-layer fully-connected network on top of the visual embeddings alone to align videos with a list of corresponding step (language) embeddings using the Drop-DTW loss, (8). Notably, the number of different step descriptions in a video is orders of magnitude smaller than the number of video clips, which leads to degenerate alignments when training with an alignment loss (i.e., most clips are matched to the most frequently occurring step). To regularize the training, we introduce an additional clustering loss, $\mathcal{L}_{\text{clust}}$, defined in the supplemental. $\mathcal{L}_{\text{clust}}$ is an order-agnostic discriminative loss that encourages video embeddings to cluster around the embeddings of steps present in the video. Finally, we use $\mathcal{L}_{\text{DTW}}$ with asymmetric costs, (4), and either a 30%-percentile drop cost, (5), or the learned variant, (6), in combination with $\mathcal{L}_{\text{clust}}$ during training.

At test time, we are given a video and the corresponding ordered list of step descriptions. To temporally localize each of the steps in the video, we align the learned vision and (pre-trained) language embeddings using Drop-DTW and directly find step boundaries.

### 4.2.1 Drop-DTW as a loss for step localization

We now compare the performance of Drop-DTW to various alignment methods [1, 5, 6], as well as previous reported results on the datasets [44, 11, 53, 54, 45, 47]. In particular, we compare to the following baselines:

- *SmoothDTW* [1]: Learns fine-grained representations in a contrastive setting using a differentiable approximation of DTW. Drop-DTW uses the same approximation, while supporting the ability to skip irrelevant parts of sequences.

- *D³TW* [5]: Uses a different differentiable formulation of DTW, as well as a discriminative loss.

- *OTAM* [6]: Extends D³TW with the ability to handle outliers strictly present around the endpoints.

- *MIL-NCE* [45]: Learns strong vision and language representations by locally aligning videos and narrations using a noise contrastive loss in a multi-instance learning setting. Notably, we learn our embeddings on top of representations trained with this loss on the HowTo100M dataset [44]. There is a slight difference in performance between our re-implementation and [44], which is due to the difference in the input clip length and frame rate.

The results in Table 1 speak decisively in favor of our approach, where we outperform all the baselines on all datasets across all the metrics with both the learned and percentile based definitions of our drop costs. Notably, although we are training a shallow fully-connected network on top of the MIL-NCE's [45] frozen pretrained visual representations, we still outperform it with sizeable margins due to the global perspective taken in aligning the two modalities. Also, note that the results of all other DTW-based methods [1, 5, 6] are on-par due to their collective inability to handle interspersed background frames during alignment, as shown in Fig. 4. This figure also motivates the use of Drop-DTW as the inference method for step localization. It uses the drop capability to 'label' irrelevant frames as background, without ever learning a background model, and does it rather accurately according to Fig. 4, which other methods are not designed to handle.

**Table 1: Step localization results on the CrossTask [9], COIN [10], and YouCook2 [11] datasets**.

| Method | | CrossTask | | | COIN | | | YouCook2 | | |
|---|---|---|---|---|---|---|---|---|---|---|
| | | Recall ↑ | Acc. ↑ | IoU ↑ | Recall ↑ | Acc. ↑ | IoU ↑ | Recall ↑ | Acc. ↑ | IoU ↑ |
| Non-DTW Methods | MaxMargin [44] | 33.6 | - | - | - | - | - | - | - | - |
| | UniVL [47] | 42.0 | - | - | - | - | - | - | - | - |
| | NN-Viterbi [54] | - | - | - | - | 21.2 | - | - | - | - |
| | ISBA [53] | - | - | - | - | 34.3 | - | - | - | - |
| | ProcNet [11] | - | - | - | - | - | - | - | - | 37.5 |
| | MIL-NCE [45] | 39.1 | 66.9 | 20.9 | 33.0 | 50.2 | 23.3 | 70.7 | 63.7 | 43.7 |
| DTW-based Methods | SmoothDTW [1] | 43.1 | 70.2 | 30.5 | 37.7 | 52.7 | 27.7 | 75.3 | 66.0 | 47.5 |
| | D³TW [5] | 43.2 | 70.4 | 30.6 | 37.9 | 52.8 | 27.7 | 75.3 | 66.4 | 47.1 |
| | OTAM [6] | 43.8 | 70.6 | 30.8 | 37.2 | 52.6 | 27.2 | 75.4 | 66.9 | 46.8 |
| | Drop-DTW + Percentile drop cost | 48.2 | **73.5** | 34.4 | 40.8 | 54.8 | **29.5** | **77.4** | 68.4 | **49.4** |
| | Drop-DTW + Learned drop cost | **49.1** | 71.3 | **34.5** | **42.8** | **59.6** | 29.5 | 76.8 | **69.6** | 48.4 |

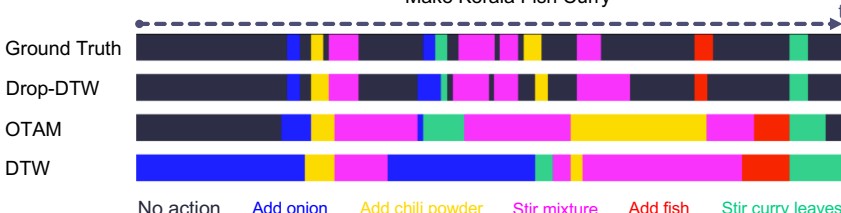

**Figure 4: Step localization with DTW variants.** Rows two to four show step assignment results when the same alignment method is used for training and inference. Drop-DTW allows to identify interspersed unlabelled clips and much more closely approximates the ground truth. More examples are provided in the supplemental.

#### 4.2.2 Drop-DTW as an inference method for step localization

Here, we further investigate the benefits of Drop-DTW (with the percentile drop cost) as an inference time algorithm for step localization. We compare to standard DTW [2] that does not allow for element dropping, OTAM [6] that allows for dropping elements around endpoints, LCSS [7] that greedily excludes potential matches by thresholding the cost matrix, prior to the alignment operation, and the Needleman-Wunsch algorithm [8] that uses a drop costs to reject outliers, but restricts the match space to one-to-one correspondences, making it impossible to match several frames to a single step.

Notably, Drop-DTW does not restrict possible matches and can infer the start and end times of each step directly *during* the alignment as our algorithm supports skipping outliers. To demonstrate the importance of performing dropping and matching within a single optimal program, we also consider the naive baseline called "greedy drop + DTW" that drops outliers *before* alignment. In this case, a clip is dropped if the cost of matching it to any step is greater than the drop cost (5). Alignment is then performed on the remaining clips using standard DTW.

In all cases, we use video and step embeddings extracted from a pre-trained network [45] to remove the impact of training on inference performance. Table 2 shows that Drop-DTW outperforms all the existing methods on all metrics and datasets by a significant margin. This is due to their inability to drop interspersed outliers or their restricted space of possible matches. More interestingly, comparing Drop-DTW to "greedy drop + DTW" reveals the importance of formulating dropping and alignment together as part of the optimization, and finding the optimal solution to (2) rather than a greedy one.

### 4.3 Drop-DTW for representation learning

In this section, we demonstrate the utility of the Drop-DTW loss, (8), for representation learning. We train the same network used in previous work [15, 1] using the alignment proxy task on PennAction [12]. Similar to previous work [15, 1], we evaluate the quality of the learned representations (i.e., embeddings) on the task of video alignment using Kendall's Tau metric. Notably, the PennAction videos are tightly temporally cropped around the actions, making it suitable for an alignment-based proxy task. To assess the robustness of Drop-DTW to the presence of outliers, we also contaminate PennAction sequences by randomly introducing $p\%$ frames from other activities to each sequence. Note that this type of data transformation is similar to a realistic scenario where a video of a baseball game sequentially captures both baseball pitch and baseball swing actions, as well as crowd shots. For various amounts of outlier frames, we contaminate PennAction videos as specified above and train a network to perform sequence alignment with the SmoothDTW [1] or Drop-DTW loss. Notably, no regularization loss is needed here and the drop cost is defined according to (5). Fig. 5 demonstrates the strong performance of Drop-DTW, which once again proves to be more resilient to outliers.

**Table 2: Comparison of different alignment algorithms for step localization as the inference procedure.** The first column describes the alignment algorithm used for inference. The video and text features used for alignment are obtained from the model [45] pre-trained on the HowTo100M dataset.

| Method | CrossTask | | COIN | | YouCook2 | |
|---|---|---|---|---|---|---|
| | Acc. ↑ | IoU ↑ | Acc.↑ | IoU ↑ | Acc.↑ | IoU ↑ |
| DTW [2] | 11.2 | 10.1 | 21.6 | 18.3 | 35.0 | 31.1 |
| OTAM [6] | 19.5 | 11.6 | 26.5 | 19.5 | 43.4 | 34.7 |
| LCSS [7] | 50.3 | 4.1 | 47.0 | 4.5 | 43.4 | 9.0 |
| Needleman-Wunsch [8] | 64.8 | 9.5 | 52.1 | 7.4 | 50.1 | 11.7 |
| greedy drop + DTW | 60.1 | 13.8 | 45.0 | 18.9 | 54.3 | 34.1 |
| Drop-DTW (ours) | **66.9** | **20.8** | **52.7** | **27.7** | **66.0** | **47.5** |

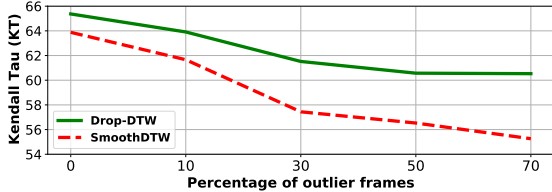

**Figure 5: Representation learning with Drop-DTW.** Video alignment results using Kendall's Tau metric on PennAction [12] with an increasing level of outliers introduced in training.

**Table 3: Unsupervised audio-visual cross-modal localization on AVE [13].** A2V: visual localization from audio segment query; V2A: audio localization from visual segment query.

| Model | A2V ↑ | V2A ↑ |
|---|---|---|
| OTAM [6] | 37.5 | 32.7 |
| SmoothDTW [1] | 39.8 | 33.9 |
| Drop-DTW (ours) | **41.1** | **35.8** |
| Supervised [13] | 44.8 | 34.8 |

### 4.4 Drop-DTW for audio-visual localization

To show the benefits of Drop-DTW for unsupervised audio-visual representation learning, we adopt the cross-modality localization task from [13]. Given a trimmed signal from one modality (e.g., audio), the goal is localize the signal in an untrimmed sequence in the other modality (e.g., visual). While previous work [13] use supervised learning with clip-wise video annotations indicating the presence (or absence) of an audio-visual event, we train our models completely *unsupervised*. We use a margin-based loss that encourages audio-visual sequence pairs from the same video to be closer in the shared embedding space than audio-visual pairs from different videos, and measure the matching cost between the sequences with a matching algorithm. When applying Drop-DTW, we use symmetric match costs, (3), and 70%-percentile drop costs, (5). Otherwise, we closely follow the experimental setup from [13] and adopt their encoder architecture and evaluation protocol; for additional details, please see the supplemental.

In Table 3, we compare different alignment methods used as a sequence matching cost to train the audio-visual model. We can see that Drop-DTW outperforms both SmoothDTW and OTAM due to its ability to drop outliers from both sequences. Interestingly, Drop-DTW outperforms the fully-supervised baseline on visual localization, given an audio query. We hypothesize that this is due to our triplet loss formulation, that introduces learning signals from non-matching audio-visual signals, while supervised training in [13] only considers audio-visual pairs from the same video.

### 4.5 Discussion and limitations

Even though Drop-DTW can remove arbitrary sequence elements as part of the alignment process, the final alignment is still subject to the monotonicity constraint, which is based on the assumption that the relevant signals are strictly ordered. While not relevant for the applications presented in this paper, training from partial ordered signals using Drop-DTW is not currently addressed and is an interesting subject for future research.

## 5 Conclusion

In summary, we introduced an extension to the classic DTW algorithm, which relaxes the constraints of matching endpoints of paired sequences and the continuity of the path cost. This relaxation allows our method to handle interspersed outliers during sequence alignment. The proposed algorithm is efficiently implemented as a dynamic program and naturally admits a differentiable approximation. We showed that Drop-DTW can be used both as a flexible matching cost between sequences and a loss during training. Finally, we demonstrated the strengths of Drop-DTW across a range of applications.

## Acknowledgments

Nikita Dvornik is supported by the MITACS Accelerate Fellowship. Animesh Garg is a CIFAR AI Chair and is supported in part by NSERC Discovery Grants and Early career grants.

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
