# OpenReview forum: "Drop-DTW: Aligning Common Signal Between Sequences While Dropping Outliers"
_NeurIPS.cc/2021/Conference — NeurIPS 2021 Poster_

### Official Review · Reviewer_wLu7 · 2021-06-28

**Rating:** 6
**Confidence:** 4

**Summary:**

The paper presents a variant of DTW, denoted as Drop-DTW.  The author introduce drop costs, which is a criteria for deciding whether or not to ignore frames in times series. By ignoring some frames in input times series, the alignment performance between times series containing non-corresponding frames can be improved. They also propose a differentiable version using the softmin function for use as a loss function. Several experiments (e.g., video retrieval, weakly supervised action labeling, and representation learning) demonstrate the superiority of the proposed method over existing method.

**Limitations And Societal Impact:**

The limitation is clarified in the main text.

**Main Review:**

# After Rebuttal

First, I would like to thank the authors for their detailed rebuttal and discussion, including additional experiments.

My main concern was that the idea of incorporating the drop technique into DP is very common in the bioinformatics field.
In response to the concern, the authors have shown the difference in performance with NW by additional experiments. In addition, the deep discussions with reviewer VyHz showed that the "one-to-many matching" of DTW has an important role in their task.
I think that their propose i.e.,  incorporating the drop technique into DTW has some values for our community, although the idea of drop+DP is quite common.
I have slightly increased my score to reflect this.

# Initial Review

## Pros.
1. In actual tasks e.g., weakly supervised action labeling and video alignment, there are many frames that do not correspond. The authors propose a reasonable solution to the problem by introducing the drop costs in the DP of the DTW.
1.  Various experiments demonstrate the superiority of the proposed method.

## Cons.
1. The drop cost itself is very common in the field of bioinformatics, while the paper does not mention it.  The major algorithms (e.g., Needleman-Wunsch[1] and Smith-Waterman[2]) and some minors (e.g., differentiable variant [3]) seem to be very related to this work. I would like the authors to clarify the relationship.
1. The drop costs are set to various values in the experiments, and some of them (Sec. 4.1.1) seem to be fixed without the percentile contrary to the claim in Sec. 3.3. I would like the authors to clarify how these values were determined. It would be even better if the impact of drop costs on performance could be investigated.
    - On a related note, please clarify the criteria for dropping in "drop then DTW" setting. Is it possible to achieve similar performance to drop-DTW, if we carefully set the drop criteria?


## Comments

1. I think Eq.6 should be $\text{smoothMin}(x; \gamma) = x \cdot \text{softmin}(x/\gamma) = \frac{x \cdot e^{-x/\gamma}}{\sum e^{-x/\gamma}}$ or $\text{smoothMax}(x; \gamma)=...$.
1. I could not find $\mathcal L_\text{clust}$ in the supplementary. $\mathcal L_\text{reg} = \mathcal L_\text{clust}$?
    - If so, does Table 1 show that "attention-based pooling" without time series information ($\mathcal L_\text{clust}$ in the table) is better than the vanilla Drop-DTW? Why?
    - (minor): This regularization seems to be related to canonical time warping[4] and deep canonical time warping[5].
    - Are there any hyperparameters for $\mathcal L_\text{clust}$? (i.e., the total loss is $\mathcal L_\text{DTW} + \lambda \mathcal L_\text{clust}, \text{where}\  \lambda=1$)?
1. If the drop cost is determined by the percentile of the match costs (depending on the inputs), strictly speaking, it may not be called "fully differentiable".
1. I could not find any info on licenses and computational resources in the supplementary, contrary to the checklists.


## References
- [1] S. Needleman and C. Wunsch. A general method applicable to the search for similarities in the amino acidsequence of two proteins.Journal of Molecular Biology, 48(3):443–453, 3 1970
- [2] T. F. Smith and M. S. Waterman. Identification of common molecular subsequences.Journal of molecularbiology, 147(1):195–7, Mar. 1981
- [3] Koide, Satoshi, Keisuke Kawano, and Takuro Kutsuna. "Neural edit operations for biological sequences." Advances in Neural Information Processing Systems 31 (2018): 4960-4970.
- [4] Zhou, Feng, and Fernando Torre. "Canonical time warping for alignment of human behavior." Advances in neural information processing systems 22 (2009): 2286-2294.
- [5] Trigeorgis, George, et al. "Deep canonical time warping." Proceedings of the IEEE Conference on Computer Vision and Pattern Recognition. 2016.

**Time Spent Reviewing:**

5

---

> ### Author Response · Authors · 2021-08-10
> **Response to Reviewer wLu7**
>
> >* drop cost itself is very common in the field of bioinformatics, while the paper does not mention it.
>
> We thank the reviewer for highlighting these related works. Here, we comment on significant distinctions between our work and the suggested references. These clarifications will be included in the revised manuscript.
>
> * While Needleman-Wunsch introduces a drop cost and allows to drop some elements from matching, it is much less general than Drop-DTW in terms of possible alignments. First, it does not allow multiple elements from one sequence to match to the same element in the other sequence (i.e. the algorithm is restricted to choosing a diagonal path in cases of a match and dedicates the alternative paths to insertions or deletions). This will have a *profound* negative effect on the step localization application, where the step sequence is typically an order of magnitude shorter than the video sequence. Concretely, in the video step localization task, Drop-DTW allows multiple frames to be matched to the same step, thus enabling sequence step segmentation, whereas this is not possible with Needleman-Wunch. To support these claims, here we provide comparisons to Needleman-Wunsch:
>
> |  | CrossTask | CrossTask | CrossTask |
> | --- | ----------- |---|---|
> | | Recall | Acc | IoU |
> | Needleman-Wunch | 43.8 |68.4 |29.4 |
> | Drop-DTW | **48.9** |**71.3** | **34.2**|
>
> Clearly, training with Needleman-Wunsch degrades the performance on all the metrics, compared to training with Drop-DTW.
>
> * Smith–Waterman is a variant of the Needlman-Wunch algorithm, which further allows for sub-sequence matching (i.e., it further relaxes the matching endpoints constraints). In contrast, our method relaxes the matching endpoints constraints using the same self-contained algorithm without requiring any additional considerations. In addition, Smith–Waterman relies on resetting negative matching costs to 0 to promote local matching, which makes the method less applicable for feature learning as it is not clear how to implement a differentiable version of this method.
> ***
> >* The drop costs are set to various values in the experiments...
>
> When training representations with Drop-DTW, the matching costs obtained by the model evolve during the training. To have the drop cost change accordingly, we define it as a percentile of the matching costs. Here, we provide results of setting the drop cost to various percentile values. In addition, we also present an alternative drop cost definition, where we represent a drop cost as a function (parametrized by a 1-layer MLP) that takes sequence x and the average step representation z as input, and produces a vector of drop costs as output (one drop cost for each element in x):
>
> |  | CrossTask | CrossTask | CrossTask |
> | --- | ----------- |---|---|
> |Drop-DTW percentile | Recall | Acc | IoU |
> | p = 0.1 | 47.5 | 72.4 | 34.9 |
> | p = 0.3 (in paper)| 48.9 |71.3 | 34.2|
> | p = 0.5 | 46.2 | 69.5|31.0 |
> | p = 0.7 | 45.3 | 68.4 | 31.0 |
> | p = 0.9 |  44.6 |67.7 |30.0 |
> | learned drop-cost | **49.7** |**74.1** |**36.9** |
>
> The results provided here demonstrate the robustness of the proposed Drop-DTW algorithm for various choices of the percentile. Interestingly, this simple instantiation of a learned drop-cost yields the best overall performance, which demonstrates the adaptability of the Drop-DTW algorithm and potential for future work.
>
> In the TMNIST experiment, we use a fixed pre-trained model (i.e., no training is involved), which allows us to find a constant threshold (using cross-validation) that reasonably separates positive and negative matches.
>
> >* please clarify the criteria for dropping in "drop then DTW" setting.
>
> We drop an element from a sequence if its drop cost is lower than any alternative matching cost.
>
> >* Is it possible to achieve similar performance to drop-DTW, if we carefully set the drop criteria?
>
> The same drop cost used for Drop-DTW is used for drop-**then**-DTW in all our experiments. As shown in the table below, the best-overall results with drop-**then**-DTW (at p=0.3) were already reported in the paper. We will include the following ablation, which demonstrates that Drop-DTW outperforms drop-then-DTW for all choices of the drop percentile:
>
> |  | CrossTask | CrossTask |
> | - | -------- | ----------- |
> |drop-**then**-DTW percentile | Acc | IoU |
> | p = 0.1 | 67.1    | 14.7  |
> | p = 0.3 (in paper) | 66.2    | 26.2|
> | p = 0.5 | 59.2    | 24.0 |
> | p = 0.7 | 33.6    | 21.1 |
> | p = 0.9 | 20.1    | 18.2 |
> | _Drop-DTW (p=0.3)_ (in paper) | _71.3_    | _34.2_ |
>
>
> >* does Table 1 show that "attention-based pooling" without time series information ( $L_{clust}$ in the table) is better than the vanilla Drop-DTW? Why?}
>
> In the instructional video step localization experiment, training with only a sequence alignment loss of any form (i.e. Drop-DTW, softDTW, D3TW or OTAM) leads to a degenerate solution. This is not specific to our method but rather to the instructional video problem setup, i.e. aligning long (\~200 elements) video sequences to short (\~10 items) step sequences allows for degenerate solutions. However, when the training is regularized by using $L_{clust}$ as an additional loss, Drop-DTW shows a clear advantage over the other sequence alignment methods under **all** the metrics (see Table 3). This suggests that the contribution of Drop-DTW is significant. Importantly, note that for other experiments on real datasets that do not suffer from the same imbalance (e.g. Exp 4.2 and 4.3), Drop-DTW again shows superior performance without reliance on the regularization loss. To clarify this point, we will move the cluster loss to the main manuscript and include this corresponding discussion.
>
>
> >* If the drop cost is determined by the percentile of the match costs (depending on the inputs), strictly speaking, it may not be called "fully differentiable".
>
> The Drop-DTW algorithm is fully-differentiable with respect to the model's parameters, however is not differentiable with respect to the hyperparameters. In this instantiation, the drop-cost can be seen as a hyperparameter and therefore, the method is differentiable. However, a learned drop-cost can also be considered as discussed above. In this case, the method becomes fully-differentiable, including the drop cost.
>
> >* I could not find any info on licenses and computational resources in the supplementary, contrary to the checklists... equation 6 should be...
>
> Thanks for highlighting these issues and typos. All our experiments were run using a single V100 GPU. Because we start from feature representations, our training/inference procedure is done in less than 2 hours. Equation 6 is indeed missing minus signs and will be fixed in the revised manuscript. As for the license, we are using publically available datasets with one of the following license types: Creative Commons Attribution-Share Alike 3.0 (MNIST), MIT (YouCook2), or BSD-3 (CrossTask), dataset-specific license (COIN), or no license (PennAction, AVE).
>
> >* This regularization seems to be related to canonical time warping[4] and deep canonical time warping[5].
>
> In Canonical Time Warping, the goal of the CCA step is to bring the representations of the two modalities into a common embedding space to facilitate the alignment via DTW. In our case, this projection corresponds to using a neural network and DTW variant as a loss function to learn a strong representation via alignment (an alignment that supports dropping **interspersed** outliers, unlike standard DTW). Despite these differences, one can indeed draw parallels between CCA and our $L_{clust}$ since the goal of $L_{clust}$ is also to promote more coherent representations between sequences.

---

### Official Review · Reviewer_6MvG · 2021-07-06

**Rating:** 8
**Confidence:** 4

**Summary:**

I really enjoyed reading this paper. It was very well written and positioned against existing works, and the technical aspects explained clearly. I have a few concerns but overall it would make for an excellent inclusion into the NeurIPS program.

The paper addresses the problem of aligning two contiguous temporal sequences, e.g., an untrimmed video and a given activity to match. Unlike previous approaches that make use of dynamic time warping to align the sequences, this paper proposes to model and remove outliers from the matching process. This could be done as a two-stage process: first find and remove outlier elements and then, second, apply dynamic time warping to the remaining sub-sequences. Rather than doing this the paper formulates a joint objective that aligns inliers (enforcing the monotonicity constraint) while allowing outliers to not be matched (i.e., not incur a mismatch penalty).

The approach is differentiable and is used as the loss function of a deep learning model. Experiments show that the method outperforms previous approaches on some standard tasks. In particular it does better than the two-stage baseline of first removing outliers and then matching as two disjoint steps.

**Ethical Concerns:**

None.

**Limitations And Societal Impact:**

Yes.

**Main Review:**

As mentioned above the paper is very well written. While reading the introduction I was slightly confused thinking that the paper was proposing a two-stage solution rather than a joint (modified DTW) objective (see lines 37--41). However, later it is clear that the detection and removal of outliers is integrated into a single objective, which is then made differentiable.

Regarding the latter, I was not completely convinced that the min operator needs to be approximated by smoothMin in Eqn. 6 for the DTW loss function to be differentiable (or at least subdifferentiable). Could the (exact) min not be retained and a subgradient computed for the optimal assignment? This would be faster and remove one hyperparameter from consideration.

My other bigger concern relates to the sensitivity of the trade-off between the "drop costs" and matching costs. It seems that if these are not carefully set then the model could learn that all elements are outliers and produce a trivial M^\star = 0 solution, or not as extreme but equally problematic prefer very short sequences where matching scores are very high. This doesn't appear to have been a problem in the solutions, but I'd like the authors to still comment on whether its a potential issue that may bias the solutions. Specifically, how sensitive are the results to the choice of s in Eqn. 3? Moreover, the choice of d^x and d^z in Section 4.1.1 does not follow the same rule for assigning drop costs.

Overall the paper makes a strong contribution. Clarification (ablation analysis) on the above-mentioned concerns would help to strengthen the paper even further.



**Time Spent Reviewing:**

3

---

> ### Author Response · Authors · 2021-08-10
> **Response to Reviewer 6MvG**
>
> >* ...Could the (exact) min not be retained and a subgradient computed for the optimal assignment?...}
>
> We thank the reviewer for the interesting suggestion. We implement the training with hard min and show the results below, in comparison to SmoothMin used in the paper.
>
> |  | CrossTask | CrossTask | CrossTask |
> | --- | ----------- |---|---|
> | | Recall | Acc | IoU |
> | SmoothMin [1] (in paper)| 48.9 |71.3|34.2|
> | Hard min | 48.3 |71.2|34.3|
>
> The results show that it is possible to use the original min operator for training and to obtain comparable performance to the differentiable min approximation.
> ***
> >* My other bigger concern relates to the sensitivity of the trade-off between the "drop costs" and matching costs. It seems that if these are not carefully set then the model could learn that all elements are outliers and produce a trivial solution.
>
> As long as we set the drop cost as a percentile of all the matching costs, we ensure that the drop cost is somewhere between the min and max matching cost. Subsequently, this ensures that some (but not all) elements will be dropped. Here, we also provide an ablation on the effect of varying the percentile choice on the final results:
>
> |  | CrossTask | CrossTask | CrossTask |
> | --- | ----------- |---|---|
> |Drop-DTW percentile | Recall | Acc | IoU |
> | p = 0.1 | 47.5 | 72.4 | 34.9 |
> | p = 0.3 (in paper)| **48.9** |**71.3** | **34.2**|
> | p = 0.5 | 46.2 | 69.5|31.0 |
> | p = 0.7 | 45.3 | 68.4 | 31.0 |
> | p = 0.9 |  44.6 |67.7 |30.0 |
>
> The results provided here demonstrate that Drop-DTW is rather robust to the choice of p. Importantly, as mentioned on lines 135-136, we argue that our percentile based definition is advantageous as it allows for an intuitive method to define the drop cost based on the expected amount of outliers present in the sequences.

---

> > ### Comment · Reviewer_6MvG · 2021-08-23
> > **Acknowledgement**
> >
> > Thank you for addressing my concerns. I am still in support of this paper.

---

### Official Review · Reviewer_VyHZ · 2021-07-16

**Rating:** 5
**Confidence:** 4

**Summary:**

Paper proposes an approach for a sequence-to-sequence alignment specifically designed to handle sequences with outliers. The core of the approach is a simple extension to the DTW where "no match" is allowed and carries a certain cost. The standard DTW can then solve for the matching path, while selecting outliers (elements matched to "no match" token). This DTW variant is converted into a loss by following for mutation in [Hadji et al] and then shown on a number of tasks where it performs well.

**Ethical Concerns:**

None.

**Limitations And Societal Impact:**

Yes, no negative impacts need to be considered.

**Main Review:**

The paper is well written and is easy to follow (though some details are relegated to supplemental materials which makes the flow a bit disjointed at times). The results are also competitive and compelling.

The real issue is the novelty. The approach relies heavily on [Hadji et al]. The core technical novelty is the extension of DTW to allow outlier dropping. However, this is not really novel. Authors claim that prior DTW variants "enforce correspondences between all elements of both sequences", a limitation that the paper is trying to solve. While this is indeed the most typical setup of DTW, other variants exist that do not require such assumptions. For example, subsequence variant of the DTW algorithm which allows partial matching also exists (e.g., see Meinard Muller. Information Retrieval for Music and Motion. Springer Verlag, 2007. or description of LCSS in https://www.cs.unm.edu/~mueen/DTW.pdf). As such the novelty of the approach is fairly marginal. In the very least, the paper should discuss and compare against such variants.

I think the presented approach is useful and has not been illustrated in the context of problems being considered. But the novelty, in my opinion, is clearly below what one would expect from a NeurIPS paper. I think the paper maybe better suited for a lower-tier venue.

Minor comments:
- "One of the main downsides of extant DTW-based approaches is that they require clean, tightly cropped, data with matching endpoints." This is not true of all DTW approaches. There are specific variants that do not require matching endpoints. Further, some do deal with noisy observation (e.g., Generalized Canonical Time Warping) at least in the feature space.

- For Experiments in Figure 3, how do you explain higher performance of Drop-DTW at 0 noise?

- For Experiments in Figure 3, do you assume that amount of noise is know for each operating point? Alternatively, how is the drop cost set? This doesn't seem to be discussed.

--- Post Rebuttal ---

In light of the lengthy discussion with the authors, I am raising my score. This new score is a compromise between three factors: (a) acknowledging that I now do believe the approach has fundamental merits and the new experimental results provided in the discussion are highly valuable and illustrative, (b) the novelty, while I no longer perceive as marginal, is still moderate; and (c) the **very** significant amount of work and revisions necessary (in my opinion) to be carried out by the authors for the camera ready. Hence, while I no longer believe the work is unacceptable for publication at NeurIPS, and would go along with the acceptance decision, I also cannot bring myself to strongly argue for the acceptance itself, largely due to (c). As such, I am effectively converging to a Borderline-ish rating and will continue to engage in discussions with other Reviewers and AE to arrive at a final collective decision.

On a personal note, I do want to thank the authors for engaging in a continued and valuable discussion.

**Time Spent Reviewing:**

6

---

> ### Author Response · Authors · 2021-08-10
> **Response to Reviewer VyHZ**
>
> >* The approach relies heavily on [Hadji et al].
>
> We only use the SmoothMin approximation of the min operator proposed in \[Hadji et al\] (i.e. ref [1] in our paper) to enable differentiability. This is not a key component of our contribution and other differentiable min operators such as SoftMin \[Cuturi et al\] (i.e. ref [4] in our paper) or the standard (hard) min operator can be used as well. The table below displays the performance of Drop-DTW on step localization, when used with different min approximations for training:
>
> |  | CrossTask | CrossTask | CrossTask |
> | --- | ----------- |---|---|
> | | Recall | Acc | IoU |
> | Drop-DTW + SoftMin [4] | 46.1 | 70.8|32.9|
> | Drop-DTW + SmoothMin [1](in paper) | 48.9 |71.3|34.2|
> | Drop-DTW + Hard min | 48.3 |71.2|34.3|
> | _[Hadji et al]_ | _43.1_ |_70.2_|_30.5_|
>
> It is clear from this table that Drop-DTW, with several different min operators (Rows 1-3), provides superior results to [Hadji et al] (Row 4). This highlights the significance of our contribution and the lack of reliance on [Hadji et al].
>
> >* The core technical novelty is the extension of DTW to allow outlier dropping. However, this is not really novel... For example, a subsequence variant of the DTW algorithm which allows partial matching also exists (e.g., see Meinard Muller. Information Retrieval for Music and Motion. Springer Verlag, 2007. or description of LCSS in https://www.cs.unm.edu/~mueen/DTW.pdf).
>
> We thank the reviewer for pointing out these additional references, which we will include and discuss in our revised manuscript. Here, we clarify the distinctions with each reference:
>
> * Meinard Muller discusses an approach that allows for dropping outliers **only** around endpoints. This approach is *exactly* the one used most recently in [6] and is referred to as OTAM in our paper.
>
>   In comparison, our method is more general as we support dropping **interspersed** outliers including those around endpoints, thereby automatically supporting sub-sequence matching using the same unified algorithm. We discuss this distinction in lines 72-74. Note that we include a direct comparison to [6]/OTAM (denoted as OTAM) in Tables 2 and 3 as well as Figure 4, with Drop-DTW showing significant improvements.
>
>
> * While LCSS allows skipping some elements from matching, there are 3 important distinctions with Drop-DTW:
>   1. the notion of drop cost is lacking;
>   2. the decision on whether to drop or keep a match is based on thresholding the similarity matrix using an "if statement" which is not differentiable;
>   3. thresholding is a greedy operation that leads to a smaller space of possible alignments, i.e. if a match is thresholded, it can not be recovered even if it leads to a better global solution.
>
>   In contrast, Drop-DTW is fully differentiable and considers all the possible match pairs in the alignment computation, which allows Drop-DTW to find the optimal alignment, given the drop costs. In fact, LCSS, is closely related to what we call drop-*then*-DTW and comparison to such approach is provided in Table 1 of our paper, where we show that our Drop-DTW algorithm outperforms the naive approach of thresholding first and then aligning using DTW (i.e. drop-*then*-DTW).
>
>   In addition, we also performed a direct comparison between LCSS and Drop-DTW for step localization inference, which shows that Drop-DTW is indeed superior (i.e. CrossTask: Ours \{Acc: **71.3**, IoU: **34.2**\} vs. LCSS \{Acc: 66.1, IoU: 23.7\}).
>
>
> * Generalized Canonical Time Warping (GCTW) alternates between solving the temporal alignment using standard DTW, and computing linear projections of data from different modalities using Canonical Correlation Analysis (CCA). Concretely, this means that the goal of CCA is to bring the representations of the two modalities into a common embedding space to facilitate the alignment. In contrast, we use a neural network to nonlinearly embed the two modalities and our Drop-DTW function as a loss to find a good representation that supports optimal alignment, while dropping outliers in a unified framework. Also, GCTW uses CCA to project the representation to a lower-dimensional feature space thereby dealing with noise in the *feature* space to some extent but is incapable of dropping portions of the inputs that coincide with outliers in the input sequences. In contrast, by design, our method handles noise at the semantic level (i.e. dropping portions of the video that do not have a match in another modality, such as step descriptions.)
>
> >* For Experiments in Figure 3, how do you explain the higher performance of Drop-DTW at 0 noise?
>
> As mentioned on lines 168-169, for this experiment we match TMNIST-part (which contains partial trajectory executions) to TMNIST-full (which contains the full trajectories). Therefore, even under 0% noise, an optimal alignment should be able to drop outliers around endpoints (i.e. perform subsequence matching). At 0% noise, Drop-DTW shows better results because it drops the start and/or end elements from TMNIST-full sequences, that are not present in the corresponding TMNIST-part sequence. In contrast, standard DTW does not have this capability. Please see the supplementary video for a visualization (from 00:35 to 00:50).
>
> >* For Experiments in Figure 3, do you assume that amount of noise is known for each operating point? Alternatively, how is the drop cost set? This doesn't seem to be discussed.
>
> The noise level is unknown in all experiments in Figure 3. As stated on line 175, the drop cost is set to 0.3, based on the validation data. The drop cost is fixed here as we rely on features pre-trained on MNIST digit recognition (i.e. no further training is involved in this experiment). Notably, for all other experiments, the drop cost is set based on the percentile criterion defined in Equation 4.

---

### Official Review · Reviewer_n8aa · 2021-07-23

**Rating:** 6
**Confidence:** 4

**Summary:**

This paper proposed an algorithm named Drop-DTW, which enables DTW to drop outliers from the sequence-to-sequence matching. A differentiable Drop-DTW is trained for temporal step localization tasks, representation learning, audio-visual retrieval and localization. Drop-DTW gets good performance on these tasks.

**Ethical Concerns:**

There is no major ethical concern.

**Limitations And Societal Impact:**

To improve clarity,

* Table 1 needs more explanation for the right two columns ‘Drop-DTW’ and ‘drop then DTW’, maybe giving different names to differentiate these evaluation methods with the ‘Drop-DTW’ method in the bottom row?
* The L_clust loss becomes L_reg in the supplementary material.

As shown in Algorithm 1, when using a fixed value of drop cost (i.e. d=0.3), the (non-differentiable) Drop-DTW loss is equivalent to first thresholding the cost matrix then applying DTW. But in Table 1, there is a clear gap between the ‘Drop-DTW’ column and ‘drop then DTW’ column. Could you point out the reason?

The experiments show the effect of clustering loss is great on natural datasets. Maybe consider introducing the auxiliary clustering loss as one of the contributions as well.


**Main Review:**

The main technical contribution of the paper is a DTW algorithm that can drop outliers. The key modification is that the cost matrix is threshold by a fixed scalar, then only low-cost points are considered in DTW. The idea is new, but there is a concurrent paper aiming at the same purpose worth some discussion [1].

The Drop-DTW is evaluated on synthetic image sequences, and the result shows that Drop-DTW can handle synthetic sequence outliers better than vanilla DTW. When trained on real instructional video datasets, the experiments show the auxiliary clustering loss is essential to get a good performance.

The clarity of the paper can be improved. Table 1 is not clear, Figure 3 can have more legends on the matrix. Some notation should be consistent.

The paper shows a modified version of DTW loss that will be useful on practical noisy data. It gets good results on downstream tasks, but from the experimental results, I think the auxiliary clustering loss is also important, which makes me think the significance of the proposed Drop-DTW is not strong.

[1] Learning to Segment Actions from Visual and Language Instructions via Differentiable Weak Sequence Alignment. Shen et al.

# post-rebuttal

Thanks for the authors' response. I read all reviews and responses.

My main concerns were 1) confusion between Drop-DTW and Drop-then-DTW and 2) effect of $L_{\text{clust}}$. The authors have addressed these concerns in both the common response and specific response. The discussion with the reviewer VyHz clarifies the distinction between the proposed Drop-DTW and relevant prior works (MVM and Needleman-Wunch), and the authors promise to include these clarifications and revise the over-claimed core contribution.

Based on these points I decided to raise my final rating.



**Time Spent Reviewing:**

4

---

> ### Author Response · Authors · 2021-08-10
> **Response to Reviewer n8aa**
>
> >* ...The main technical contribution of the paper is a DTW algorithm that can drop outliers. The key modification is that the cost matrix is threshold by a fixed scalar, then only low-cost points are considered in DTW....
>
> As mentioned in the common response to all reviewers, Drop-DTW algorithm does **not** threshold the cost matrix by a fixed scalar; instead, it considers the original cost matrix for all the computations. The decision to drop an element is not taken greedily based on whether the drop cost is larger than the matching cost. Instead, the elements to be dropped are defined by solving the dynamic program in Algorithm 1.
>
> >* The idea is new, but there is a concurrent paper aiming at the same purpose worth some discussion [1].
>
> Note that this work appeared on the CVPR'21 website **after** our submission, and it was therefore impossible for us to include it in our paper at submission time.  While, we both support dropping outliers, note that we propose a different and more generic approach for dropping outliers. Differently from Drop-DTW, [1] approaches the muti-step localization in 2 independent steps: **1)** it converts videos and narrations into shorter sequences of prototypes and uses their proposed DTW variant to learn a representation of step prototypes; then **2)** it uses the distance between each frame and step prototypes to segment a video. In contrast, we use Drop-DTW to directly align video content to step descriptions and automatically find an optimal segmentation of the video using dynamic programming. In short, segmentation is a direct by-product of our DTW-based learned alignment, whereas in [1] it is used to learn a representation of the key steps and segmentation is obtained in a separate step. Notably, our approach yields superior performance on the same dataset (i.e. CrossTask: Ours \{recall: **48.9**, Acc: **71.3**\} vs. [1] \{recall: 35.46, Acc: 40.99\}), although a direct comparison is not entirely fair as they start from narrations on the language side.
>
> >* I think the auxiliary clustering loss is also important, which makes me think the significance of the proposed Drop-DTW is not strong.
>
> In the instructional video step localization experiment, training with only a sequence alignment loss of any form (i.e. Drop-DTW, softDTW, D3TW or OTAM) leads to a degenerate solution. This is not specific to our method but rather to the problem setup, i.e. aligning long (\~200 elements) video sequences to short (\~10 items) step sequences allows for degenerate solutions, as discussed in lines 217-222. However, when the training is regularized by using $L_{clust}$ as an additional loss, Drop-DTW shows a clear advantage over the other sequence alignment methods under **all** the metrics (see Table 3). This suggests that the contribution of Drop-DTW is significant. Importantly, note that for other experiments on real datasets that do not suffer from the same imbalance (e.g. Exp 4.2 and 4.3), Drop-DTW again shows superior performance without reliance on the regularization loss. To clarify this point, we will move the clustering loss definition to the main manuscript and include the corresponding discussion.
>
> >* As shown in Algorithm 1, when using a fixed value of drop cost (i.e. d=0.3), the (non-differentiable) Drop-DTW loss is equivalent to first thresholding the cost matrix then applying DTW. But in Table 1, there is a clear gap between the ‘Drop-DTW’ column and ‘drop then DTW’ column. Could you point out the reason?
>
> As clarified in the response to all reviewers, Drop-DTW solves the alignment and dropping problem in a **unified** framework that is fundamentally different from the naive approach of first thresholding the cost matrix and then aligning the remaining elements. In the latter case, neither the drop elements nor the alignment are optimal, which yields worse results, as shown in Table 1 of our paper.
>
> Regarding the differentiability of Drop-DTW, the algorithm is always differentiable with respect to the model's parameters, however it is not differentiable with respect to the hyperparameters, such as the percentile (or the value) of the drop cost.
>
>
> >* The clarity of the paper can be improved. Table 1 is not clear, Figure 3 can have more legends on the matrix. Some notation should be consistent
>
> We thank the reviewer for pointing out clarification points, which we will address as suggested in the revised manuscript.

---

### Author Response · Authors · 2021-08-10
**Common response**

We thank the reviewers for their positive comments on our work. We are pleased they found *"the idea new"* (R-n8aa) and *"reasonable"* (R-wLu7), the paper *"well written"* (R-VyHZ, R-6MvG), the *"technical aspects explained clearly"* (R-6MvG), and the results *"competitive and compelling"* (R-VyHZ, R-n8aa, R-wLu7, R-6MvG).
***

### We note a confusion on the difference between our proposed approach (Drop-DTW) versus a naive baseline that we call drop-**then**-DTW in our paper. Here, we first make the distinction clear.

drop-**then**-DTW addresses the problem of aligning sequences with outliers in two steps:
- **(1)** it identifies and drops the outliers using a drop-cost threshold on the matching costs, and then
- **(2)** aligns the remaining sequence elements with *standard* DTW.

Importantly, step **(1)** is done **independently** from step **(2)**, which results in an approximate greedy solution to the problem of optimal sequence matching with outliers (defined in Equation 2). Critically, if an important element is erroneously dropped in step **(1)** it is impossible to recover in step **(2)**. Moreover, the outlier rejection in step **(1)** is order-agnostic and results in drops broadly scattered over the entire sequence, which makes drop-**then**-DTW inapplicable for precise step localization.

In contrast, **Drop-DTW is a unified framework** that solves for the optimal temporal alignment while **simultaneously** detecting outliers (as noted by Reviewer R-6MvG). The drop-cost in Drop-DTW specifies the cost of dropping a frame within this optimization process. Specifically, as defined in Algorithm 1 (lines 8-10), Drop-DTW takes into account **all** potential matches with their original costs, and enables element dropping by adding another dimension to the dynamic programming table. In other words, Drop-DTW efficiently considers *all* the possible sub-sequence alignments and finds the **exact optimal solution** to the problem of sequence matching with outliers (defined in Equation 2).

**We will incorporate the above clarification in the revised manuscript.**
***
### Multiple reviewers suggested a number of existing approaches that may resemble Drop-DTW. While these approaches are relevant on the surface. However, none of them (or even their combinations) offer the same functionality as Drop-DTW. Here, we briefly highlight the key differences between these methods and Drop-DTW.
- The LCSS algorithm (mentioned by R-VyHZ) is similar to drop-**then**-DTW, i.e. it drops the outliers greedily and independently from the alignment procedure which provides poor results, as shown in Table 1 of our paper.
- The Needleman-Wunsch and Smith-Waterman algorithms (mentioned by R-wLu7) also use the idea of a drop cost. However, it implements a  domain-specific, restricted form of matches and drops, i.e., only one-to-one matches and individual drops are allowed. As a consequence, it yields sub-optimal results in our experiments (see results and detailed discussion in response to R-VyHZ). In contrast, it is important for our applications that each item can be matched with multiple items, and both individual items and pairs of items can be dropped. Our Drop-DTW implements such possibilities and provides a more general matching algorithm.
- Finally, the Canonical Time Warping (CTW) or its deep counterpart (mentioned by R-VyHZ, R-wLu7) can handle noisy *representations* by learning a robust feature transformation, however, it cannot handle outliers in sequences (i.e. it does *not* support dropping). In contrast to Drop-DTW, CTW does not modify the sequence matching algorithm to drop the outliers and uses classical DTW.

In summary, Drop-DTW is more general and powerful than existing sequence alignment algorithms, as we show below through algorithm analysis and additional experiments in response to individual reviewer comments.

**We will create a separate subsection in our related work that discusses the aforementioned works.**

---

### Author Response · Authors · 2021-08-16
**Additional feedback from Reviewers**

Dear Reviewers and Area Chairs,

The concerns about the paper raised by the reviewers were mainly clarifications on the method and related work, rather than additional experimental validation or the technicality of the research itself. We have hopefully answered all the questions in the rebuttal and will update the draft accordingly for improved comprehension.

In light of this, we request additional feedback and justification if the reviewers choose to keep their ratings.

---

> ### Author Response · Authors · 2021-08-29
> **Request for discussion**
>
> We have added additional experiments, point-wise clarifications to questions, as well as discussions about related works.
> At this point, we would deeply appreciate additional feedback and justification for current ratings.
>
> As the discussion period draws to an end, we would like to sincerely request the reviewers: [n8aa](https://openreview.net/forum?id=A_Aeb-XLozL&noteId=nIrHxikN-1r),  [VyHz](https://openreview.net/forum?id=A_Aeb-XLozL&noteId=7YFn7CxE98), and [wLu7](https://openreview.net/forum?id=A_Aeb-XLozL&noteId=ggZAX4xgGa) to kindly acknowledge the rebuttal, and point out if any additional clarifications would be useful in re-evaluation of scores.
>
> Your comments and feedback have helped us improve the paper, and we hope that you will find the improvements sufficient to corroborate the readiness of this work.

---

> > ### Comment · Reviewer_VyHZ · 2021-08-29
> > **Rebuttal response from VyHZ**
> >
> > I would like to acknowledge and thank the authors for their rebuttal.
> >
> > My comments, while indeed do not question the technical validity of the method, are in my opinion no less serious or valid and justify the current score. Just because something is technically correct, does not necessarily mean it is publishable, particularly in a top-tier ML venue. Additional criteria needs to also be assessed, including novelty, significance, validation, positioning, ect. Along those later axes is where I think most of my and other reviewer concerns are. While I appreciate additional experiments and discussion provided by the authors, I remain largely unconvinced overall.
> >
> > With regards to the rebuttal, I would like to make the following observations. I will focus here on the points that specifically address the comments from my review.
> >
> > > "We only use the SmoothMin approximation of the min operator proposed in [Hadji et al] (i.e. ref [1] in our paper) to enable differentiability."
> >
> > Indeed, but isn't the differentiability exactly the main claim to novelty that the  approach is pushing (see response to LCSS, point 2). So the fact that you are relying on [1] for differentiability, goes directly against your claims of novelty. The argument I am making isn't specific to any form of Min function, it is with regards to the novelty of making that function differentiable. To my understanding, the proposed approach does not provide a new formulation for differentiability needed to make the method work, rather it relies on such formulation from [1].
> >
> > > "This highlights the significance of our contribution and the lack of reliance on [Hadji et al]."
> >
> > Improved performance does not equal significance of contributions :) Do not get me wrong, improved performance is very important, but in itself it is not justification of novelty or significance.
> >
> > > Novelty with respect to LCSS.
> >
> > This is a very important issue that I would like to discuss point-by-point with respect to the author's response. Specifically, authors make three novelty claims:
> >
> > (1) the notion of drop cost is lacking
> >
> > As mentioned by wLu7 the drop cost is common in practice. So I am having a hard time viewing this as significant novelty.
> >
> > (2) decision on whether to drop or keep is not differentiable
> >
> > Indeed, however, the differentiability is addressed in [1], so the use of it in context of LCSS-type of formulation is fairly straightforward. From the paper itself: "To make the matching process differentiable, we replace the hard min operator with the following differentiable approximation introduced in [1]." In other words, the solution in [1] is leveraged to address this, and hence this is also hard to view as a significant technical contribution.
> >
> > (3)  greedy thresholding
> >
> > True, but again, I think this is simply a consequence of using drop-cost and differentiable formulation. So a direct consequence of (1) and (2) rather than a separate contribution on its own.
> >
> > > Direct comparison with LCSS.
> >
> > This is a useful experiment, however, it is not clear exactly how this comparison was conducted. For example, was the drop cost set the same in both cases (e.g., using cross-validation for LCSS), how LCSS was implemented, etc. Some additional details would be much welcomed here; otherwise it is hard to make sense of these numbers.
> >
> > > Final thoughts
> >
> > The rebuttal does address some of my concerns and the additional experiments are helpful (so slight net positive). That being said, I still view the novelty as incremental and the paper as needing a substantial revision to clarify the core contributions/novelty and their significance. In particular, the original claims of novelty in the submission are at least exaggerated. Specifically, from the rebuttal, as authors discuss novelty with respect to suggested past references pointed to by myself and other reviewers, the arguments made are significantly more subtle than what was claimed initially in the submission. The subtlety that I feel needs to be incorporated into the main paper, supported by more through evaluation, and ultimately re-assessed.

---

> > > ### Author Response · Authors · 2021-08-30
> > > **Response to Reviewer VyHZ**
> > >
> > > We thank the reviewer for the comments. Below, we clarify the remaining misunderstanding and address the reviewer's concerns.
> > >
> > > The main contribution of the paper is **not** introducing a greedy alignment solution amenable to outliers (as in LCSS) and combining it with a differentiable approximation of DTW (as in Hadji et al. [1]). This would indeed be trivial, as noted by the reviewer; however, this is **not** what we are doing in this work, as highlighted by Reviewer-6MvG.
> > >
> > > The main contribution of our paper is: **formulating sequence matching with outlier dropping as an optimization problem (see Equation 2) and proposing an efficient dynamic programming algorithm (see Algorithm 1) to solve it in a unified framework. This leads to superior empirical performance compared to all alternative solutions**.
> > >
> > > >*  "isn't the differentiability exactly the main claim to novelty that the approach is pushing"
> > >
> > > We do **NOT** claim that the differentiable min() approximation is what enables our contribution. We simply demonstrate that our proposed dynamic programming algorithm (i.e., Algorithm 1) can be made differentiable and that Drop-DTW can be used as a differentiable loss function. In fact, we show in the paper (e.g., experiments on TMNIST in Section 4.1) and in the rebuttal that our formulation does **not** need the differentiable formulation to work and that it yields superior results compared to other **greedy** methods, such as LCSS.
> > >
> > > Importantly, the LCSS algorithm has two non-differentiable components, i.e., **i)** the min (or max) operator, and **ii)** the thresholding operator that is implemented as an if-statement over transitions in the dynamic programming table. While the work of [1] propose a differentiable approximation of a min function and can address **i)**, to the best of our knowledge, it can not be readily applied to differentiate through thresholding in **ii)**!
> > >
> > > >* "Improved performance does not equal significance of contributions :)"
> > >
> > > We agree, but **i)** a new optimal solution like ours, which **ii)** can work with or without a differentiable component and **iii)** also significantly surpasses the state of the art on **five** realworld datasets (which are standard vision and cross-modal benchmarks), achieves the significance criterion.
> > >
> > > ### Comparison to LCSS
> > > To further ground our explanations, we refer to the point-by-point discussion of LCSS versus Drop-DTW provided by the Reviewer.
> > >
> > > 1) >"Drop cost is not novel":
> > >
> > > The drop-cost in and of itself is not what makes our work novel. The novelty of Drop-DTW is the **optimal** formulation of alignment with outlier dropping (see Equation 2) and an efficient algorithm for solving it (i.e., Algorithm 1). The existing algorithms with a drop cost (mentioned by the reviewers) either solve the problem greedily (LCSS) or restrict the space of possible matches (Needleman-Wunsch) and show poor empirical results on the problems where Drop-DTW excels. Note that we selected the **best** hyperparameters for all alternative methods.
> > >
> > > 2) >"...the differentiability is addressed in [1], so the use of it in context of LCSS-type of formulation is fairly straightforward":
> > >
> > >
> > > We disagree. As discussed above, the decision on whether to drop or keep a match in LCSS relies on an **if statement**, which will **not** become differentiable by introducing the min() approximation of [1]. In contrast, Drop-DTW does not suffer from this shortcoming.
> > >
> > >
> > > 3) >"Optimality of the drop cost is consequence of using drop-cost and differentiable formulation".
> > >
> > > We disagree as this is a mischaracterization of our work. The optimality of our solution is the consequence of solving for outlier removal and inliers alignment in a unified framework by formulating the optimization problem in Equation 2 and solving it with Algorithm 1. This is true regardless of whether we combine our method with a differentiable or hard min formulation.
> > >
> > > For example, in Secion 4.1 we applied Drop-DTW for noisy T-MNIST sequence classification with pretrained, **frozen** features. That is, we did **not** train the features end-to-end with our alignment loss and thus did not use the differentiable min approximation of [1]. To empirically compare Drop-DTW with LCSS in this setup, we repeat the experiment of Section 4.1 with LCSS (algorithm given below) and report the **best** results of LCSS on noisy T-MNIST sequence classification (depending on the percentage of outliers) in the table below:
> > >
> > > | % of noisy frames   | 0    | 20   | 50   | 70   |
> > > |---------------------|------|------|------|------|
> > > | Drop-DTW            | 92.5 | 91.2 | 85.0 | 68.7 |
> > > | LCSS                | 83.7 | 82.5 | 73.7 | 35.0 |
> > >
> > > It is clear the Drop-DTW **outperforms** LCSS (with 33% absolute difference in accuracy when 70% of frames are blurred), even without training. This is the direct consequence of formulating inlier alignment and outlier dropping in Drop-DTW as a **single** optimization problem (see Equation 2), as opposed to the greedy formulation in LCSS.
> > >
> > > ### Implementation details of LCSS
> > >
> > > Below, we elaborate on the implementation details of LCSS with drop cost (which can be directly compared to ours in Algorithm 1):
> > >
> > > 1) Pseudo-code for our implementation of the LCSS algorithm (following [A]):
> > >     ```
> > >     # INPUT: C - cost matrix [K, N], drop_cost - fixed drop cost
> > >     D <- initialize the solution table D
> > >     for i in [1, ..., K]:
> > >         for j in [1, ..., N]:
> > >             if C[i, j] > drop_cost:
> > >                 D[i, j] = D[i-1, j-1] + drop_cost
> > >             else:
> > >                 D[i, j] = min(D[i-1, j], D[i, j-1])
> > >     return D[K, N]
> > >     ```
> > >
> > > 2) We define the drop cost, d, for LCSS exactly as done with Drop-DTW, i.e., as a p-percentile of the match costs (see Section 3.3). The optimal value of the percentile, p, for LCSS is found using cross-validation. This ensures **fair comparison** to Drop-DTW, i.e., we report the **best** results that LCSS can achieve.
> > >
> > >
> > > [A] Almotairi S., and Ribeiro E. "Human Action Recognition Using Temporal Sequence Alignment", CSCI'14
> > >
> > > ## Summary
> > > In summary, we believe that our paper together with the rebuttal fulfills all "axes" laid out by the reviewer for a NeurIPS paper:
> > > * **Novelty**: We formulate sequence matching with outlier dropping as an optimization problem (see Equation 2) and propose an efficient dynamic programming algorithm (see Algorithm 1) to solve it in a unified framework. This is in contrast to baselines included in our paper and all alternative methods pointed out by the reviewers, as highlighted in our response. Our clarification will be included in the revised paper.
> > > * **Positioning**: We clearly contrasted our approach with all other baselines with detailed descriptions throughout our rebuttal. In our initial response to all reviewers, we shared verbatim the paragraph we will include in the revised version of our paper to strengthen the positioning.
> > > * **Results**: Our method achieves state-of-the-art results across **five** realworld datasets (i.e., CrossTask, COIN, YouCook 2, PennAction, and AVE), standard in the vision and cross-modal literature, and outperform all alternative alignment-based methods.
> > > * **Significance**: We demonstrate that our method is broadly applicable: (i) as a distance measure between sequences (Section 4.1), (ii) as both a training and inference time method for multi-step localization (Section 4.2), (iii) and as a loss for representation learning in a single modality (Section 4.3) or cross-modal setup (Section 4.4). Moreover, the novelty of our method combined with the strong results compared to extant methods, demonstrates the significance of our paper.

---

> > > > ### Comment · Reviewer_VyHZ · 2021-08-30
> > > > **Continuation on novelty**
> > > >
> > > > Thank you for the response. Let me try to discuss the following couple of sentence which I believe go to the core of the novelty discussion.
> > > >
> > > > > The main contribution of the paper is not introducing a greedy alignment solution amenable to outliers (as in LCSS) and combining it with a differentiable approximation of DTW (as in Hadji et al. [1]). This would indeed be trivial, as noted by the reviewer; however, this is not what we are doing in this work, as highlighted by Reviewer-6MvG.
> > > >
> > > > > The main contribution of our paper is: formulating sequence matching with outlier dropping as an optimization problem (see Equation 2) and proposing an efficient dynamic programming algorithm (see Algorithm 1) to solve it in a unified framework. This leads to superior empirical performance compared to all alternative solutions.
> > > >
> > > > I want to highlight that LCSS is not required to be greedy. In fact, most references to LCSS discuss that it is best implemented using dynamic programming, which can obtain optimal solution. As a canonical reference for this let me use the following paper:
> > > >
> > > > Latecki LJ, Megalooikonomou V, Wang Q, Yu D (2007). “An Elastic Partial Shape Matching Technique.” Pattern Recognition, 40(11), 3069–3080.
> > > >
> > > > This paper proposes an DTW-based approach for shape matching (they call it MVM) with ability to skip outliers. I quote from the referenced paper: "MVM combines the strengths of both DTW and LCSS, while overcoming their constraints. MVM computes the distance value between two sequences that are obtained from shape boundaries. It calculates the shape similarity directly based on the distances of corresponding elements, just as DTW does, but also allows the query sequence to match to only a subsequence of the target sequence, just as LCSS does. Like DTW, MVM also tries to find an optimal path including all the corresponding pairs. But MVM is able to skip outliers during the matching process and the path does not need to be consecutive."
> > > >
> > > > Like I said, I am using this as a canonical reference, but other variants exist in the literature. Further, beam search and N-best solutions can be obtained using methods such as in [a], to find more globally optimal solutions with respect to metrics not easily expressed recursively as is required by DTW and variant objectives.
> > > >
> > > > [a] N-best maximal decoders for part models
> > > > Dennis Park, Deva Ramanan
> > > > ICCV, 2012.
> > > >
> > > > So, considering this, I do not view Algorithm 1 itself as containing significant novelty. In effect, it adds an additional (skip) state with fixed cost that is than incorporated into standard DP formulation. What I do view as novel is making Algorithm 1 differentiable, which allows learning within a NN. However, here the authors rely on formulation in Hadji et al. [1]. So this, as well, I view as incremental. To be clear, I am not arguing there is NO novelty. I am just saying it is incremental and needs to be much better characterized.
> > > >
> > > > > Details of comparison to LCSS
> > > >
> > > > Thank you! I appreciate additional details provided by the authors.

---

> > > > > ### Author Response · Authors · 2021-08-30
> > > > > **Response**
> > > > >
> > > > > We thank the reviewer for the opportunity to carry on a dialog. Below, we clarify the remaining misunderstanding and highlight the difference between Drop-DTW and the methods the reviewer has now referenced.
> > > > >
> > > > > >* I want to highlight that LCSS is not required to be greedy. In fact, most references to LCSS discuss that it is best implemented using dynamic programming, which can obtain optimal solution.
> > > > >
> > > > > We agree. LCSS, as all other referenced sequence alignment methods, uses dynamic programming to find the optimal solution. However, **the outlier rejection step in LCSS is greedy**, as it is performed before alignment and it is not a part of the dynamic programming formulation. This is contrary to Drop-DTW, where both inlier alignment and outlier rejection are solved jointly in a single optimization problem (see Equation 2).
> > > > >
> > > > > >* This paper proposes an DTW-based approach for shape matching (they call it MVM) with ability to skip outliers.
> > > > >
> > > > > MVM indeed allows to skip though outliers inside the dynamic program; however, contrary to Drop-DTW, **MVM does not allow outlier rejection in both sequences** and assumes that one of the sequence does not contain outliers. This would make the application of MVM to cross-domain representation learning (see Section 4.4) impossible. Moreover, to perform outlier rejection, **MVM restricts the space of possible matches to only one-to-one matches**. This is exactly what Needleman-Wunsch algorithm does; and we compared Drop-DTW to Needleman-Wunsch in our response to Reviewer wLu7. We copy the comparison of Drop-DTW and Needleman-Wunsch for the task of step localization here for convenience.
> > > > >
> > > > > |  | CrossTask | CrossTask | CrossTask |
> > > > > | --- | ----------- |---|---|
> > > > > | | Recall | Acc | IoU |
> > > > > | Needleman-Wunch | 43.8 |68.4 |29.4 |
> > > > > | Drop-DTW | **48.9** |**71.3** | **34.2**|
> > > > >
> > > > > It is clear that Drop-DTW outperforms this family of alignment algorithms thanks to a more flexible **one-to-many** matching, which is essential to perform step localization. **These gains are only possible thanks to the Algorithm 1 formulation!** This demonstrates that Algorithm 1 is not just novel, it's essential to solve step localization via alignment.
> > > > >
> > > > > >*beam search and N-best solutions can be obtained using methods such as in [a], to find more globally optimal solutions with respect to metrics not easily expressed recursively
> > > > > >[a] N-best maximal decoders for part models Dennis Park, Deva Ramanan ICCV, 2012.
> > > > >
> > > > > The N-best paper considers the problem of matching set elements between each other. However, **this is not the problem we are solving in this paper!** We solve alignment between 2 sequences in the presence of outliers. Since the work of [a] considers a single set, rather than a pair, and is agnostic to the order of elements in the set, we do not see how one can apply [a] to the problem of sequence alignment.
> > > > >
> > > > > ## Summary
> > > > > Above, we have demonstrated that Algorithm 1 - one of our main contributions - enables step localization by alignment, has no existing alternative, and achieves state-of-the-art results. Contrary to the reviewer's (repeated) claims, we do **not** consider differentiability as one of our main contributions.  Differentiability is simply a useful consequence of Algorithm 1, which naturally admits the min approximation proposed by Hadji et al. [1] or other differentiable min approximations (see our initial response).

---

> > > > > > ### Comment · Reviewer_VyHZ · 2021-08-30
> > > > > > **Further follow up**
> > > > > >
> > > > > > I appreciate the continued discussion and careful responses from the authors. I believe we are actually converging in characterization of novelty, so this is quite helpful. Specifically, with respect to MVM and Needleman-Wunch, which are taken as samples of close prior work, the main arguments of novelty, as per last response from the authors, appear to be:
> > > > > >
> > > > > > 1. Ability of the proposed method to handle outlier rejection in both sequences (not just one)
> > > > > > 2. Ability to do one-to-many matching in the alignment (not just one-to-one)
> > > > > >
> > > > > > I actually agree with these two points. I believe this to be a fairly objective characterization of the novel properties of the proposed model.
> > > > > >
> > > > > > I do want to note that this is a narrower set of claims as compared to what is stated in the submission itself. For example, statements such as "Drop-DTW is the first to augment DTW with the ability to skip through irrelevant parts of the signal during alignment" are over-claiming and need to better reflect the properties mentioned above.
> > > > > >
> > > > > > In any case, with the above two claims in mind, the secondary question is (conceptual) importance and significance of the two novel properties mentioned above. This is where things get fundamentally subjective. In my subjective opinion, (1) does not appear to be significant. I do not immediately see why this would prevent use of the model for cross-domain representation learning, or significantly reduce performance elsewhere. Perhaps authors can elaborate on this. I could be missing something, but I think the model would still be applicable in all considered cases. It may not be as effective (work worse), but this would require an explicit experiment to validate. I do agree that (2) is potentially important. Authors attribute improved performance of Drop-DTW over Needleman-Wunch to this one-to-many matching. While this is a reasonable hypothesis, this cannot be determined conclusively from the provided experiment. The two models differ in other aspects as well. In order to prove this out conclusively, authors should disable one-to-many matching in Drop-DTW to see what effect this may have.
> > > > > >
> > > > > > In general, while I am completely satisfied with the overall performance of the proposed approach, the attribution of improvements to specific design choices and properties of the model are much less clear (especially with respect to the two claims above).

---

> > > > > > > ### Author Response · Authors · 2021-08-31
> > > > > > > **Clarifications**
> > > > > > >
> > > > > > > We sincerely thank the reviewer for acknowledging our contribution and positively receiving our response. Indeed, it seems like we are finally converging and we appreciate the opportunity for clarification, which ultimately will make our submission stronger. Here, we address the remaining concern about significance.
> > > > > > >
> > > > > > >
> > > > > > > >* I do want to note that this is a narrower set of claims as compared to what is stated in the submission itself.
> > > > > > >
> > > > > > > We will integrate our response into the revised manuscript to make the distinctions clearer and more precisely position our work.
> > > > > > > ### Significance:
> > > > > > > >* importance and significance of the two novel properties mentioned above.
> > > > > > >
> > > > > > > * **Ability of the proposed method to handle outlier rejection in both sequences**
> > > > > > >
> > > > > > > Our approach is more general/flexible. It does not require making assumptions that one of the two sequences is outlier-free, which we believe is a **significant** distinguishing aspect between our method and all the alternatives. A concrete application demonstrating the significance of this component is provided in Section 4.4 of our main paper where we show that Drop-DTW can be used in a cross-modal setting with outliers present in both the visual and audio signal (i.e., some things can be heard but not seen, while others can be seen but not heard). In that case, we allow Drop-DTW to drop outliers from both sequences (again using the same optimal, unified framework; see complete algorithm in the supplemental) and achieve superior performance to alternative alignment methods in a completely unsupervised setting.  Notably, we are even competitive with a published fully supervised approach [13].
> > > > > > >
> > > > > > > To quantitatively demonstrate the importance of Drop-DTW's ability to drop outliers from both sequences, we repeat the experiment from Section 4.4 with the version of Drop-DTW, where the drops are only allowed to one of the sequences, i.e., only audio **or** video, (dubbed Drop-DTW-oneside) and show the results in the table below:
> > > > > > >
> > > > > > > |    Method    | Drop video| Drop audio | A2V	|V2A	|
> > > > > > > |-----------|---|---|-----------|---------|
> > > > > > > |DTW	         | no | no | 39.8 | 33.9    	|
> > > > > > > |Drop-DTW-oneside| no | yes | 39.0|	33.5|
> > > > > > > |Drop-DTW-oneside| yes | no |	38.9|	33.3|
> > > > > > > |Drop-DTW (ours) | yes | yes |**41.1**	|**35.8**|
> > > > > > >
> > > > > > > Clearly, our Drop-DTW outperforms standard DTW and the Drop-DTW-oneside variants. Interestingly, both Drop-DTW-oneside variants perform even worse than  standard DTW that cannot drop elements from either sequence. We hypothesize that this follows because standard DTW allows the outliers from both sequences to match to each other, while Drop-DTW-oneside forces all the outliers from one sequence (that is not amenable to drops) to match to only inliers from the other sequence (that admits drops), which creates more erroneous inlier-to-outlier correspondences and results in a more biased training signal. In contrast to both DTW and Drop-DTW-oneside, our Drop-DTW does not have to deal with outlier matching at all, and is able to learn from inlier-to-inlier correspondences only, which is supported by the empirical results in the table.
> > > > > > >
> > > > > > > * **Ability to do one-to-many matching in the alignment**
> > > > > > >
> > > > > > > In fact, the ability to do one-to-many matching is **exactly** what enables applications such as step localization. For example, in the case of instructional videos, the video sequence is typically an order of magnitude longer than the step sequence. For example, a typical video from the CrossTask dataset could consist of 200 video clips and contain just five instructional steps. Applying an alignment algorithm with one-to-one matches only (e.g., MVM or Needleman-Wunsch) will match the five instructional steps with only five video clips (i.e., put into one-to-one correspondence), and will be forced to drop the remaining 195 video clips, even if the steps in the video span multiple clips (as typically is the case). Clearly, such a formulation can not be used to find the start and end time of steps in a video, as the start and end time simply collapse into a single point, and most of the video is labeled as "background".  In contrast, Drop-DTW allows to assign multiple video clips to a single step and results in a meaningful temporal segmentation of the video sequence. This means that Drop-DTW does not just achieve better results on the task of step localization, it **enables** to solve this task via alignment. To support these claims, we provide an additional direct comparison to Needleman-Wunsch and Drop-DTW-one-to-one (i.e., Drop-DTW that is restricted to one-to-one matches only) for step localization as an inference procedure in the table below:
> > > > > > >
> > > > > > >
> > > > > > > |           |CrossTask | COIN  | YouCook2 |
> > > > > > > |-----------|-----------|---------|-----------|
> > > > > > > |	             | IoU    	     | IoU         | IoU    |
> > > > > > > |Needleman-Wunch|		9.5|   7.4	 | 11.7	|
> > > > > > > |Drop-DTW-one-to-one|		9.7|   8.1	 | 12.0	|
> > > > > > > |Drop-DTW	|**30.5**	| **27.7**		|**47.5**|
> > > > > > >
> > > > > > > We can see that Drop-DTW is indeed significantly better suited for step localization as it improves the Intersection over Union (IoU) metric by three to four times, compared to one-to-one alignment algorithms.
> > > > > > >
> > > > > > > >* this cannot be determined conclusively from the provided experiment. The two models differ in other aspects as well. In order to prove this out conclusively, authors should disable one-to-many matching in Drop-DTW to see what effect this may have.
> > > > > > >
> > > > > > > As described and shown above with the Drop-DTW-one-to-one example, turning off the one-to-many matching has a large, negative impact on the instructional video alignment task, as expected, as the very definition of this task requires multiple clips to be matched to one step.  We therefore believe that the provided clarification and results speak decisively in favor of our approach.
> > > > > > >
> > > > > > >
> > > > > > >
> > > > > > > ### Summary
> > > > > > > In summary, our contributions go beyond simple design choices. They actually enable a broad variety of applications, as shown in the paper. With this discussion we sincerely hope our response convinced the reviewer that our submission meets the axes that they laid out for a NeurIPS paper, and hope this will convince them to raise their rating.
> > > > > > > At this point, it appears that we have addressed all the objective concerns; what remains are the subjective remarks raised in the review and responses. If you have further concerns, please share.

---

> > > > > > > > ### Comment · Reviewer_VyHZ · 2021-08-31
> > > > > > > > **Final question**
> > > > > > > >
> > > > > > > > Thank you, this latest set of experiments is **very** helpful. I just have one final clarification to ask the authors:
> > > > > > > >
> > > > > > > > > Ability to do one-to-many matching in the alignment
> > > > > > > >
> > > > > > > > The numbers reported in this latest experiment do not appear to match with the numbers reported earlier. Specifically, in earlier experiment comparing to Needleman-Wunch the reported numbers were:
> > > > > > > >
> > > > > > > > * Needleman-Wunch: 29.4
> > > > > > > > * Drop-DTW: 34.2
> > > > > > > >
> > > > > > > > and the new numbers are:
> > > > > > > >
> > > > > > > > * Needleman-Wunch: 9.5
> > > > > > > > * Drop-DTW-one-to-one: 9.7
> > > > > > > > * Drop-DTW: 30.5
> > > > > > > >
> > > > > > > > This is a **very** significant difference, particularly in performance of Needleman-Wunch, which is 3x lower. How can this be explained? Both experiments are run on CrossTask dataset with IoU metric, so what accounts for this?

---

> > > > > > > > > ### Author Response · Authors · 2021-08-31
> > > > > > > > > **Final clarifications**
> > > > > > > > >
> > > > > > > > > We sincerely thank the reviewer for their prompt responses, acknowledging our contribution, and positively receiving our responses. We are happy to clarify the outstanding question.
> > > > > > > > >
> > > > > > > > > The reason the numbers do not match is that they correspond to two different experiments as we wanted to highlight the advantage of Drop-DTW over Needleman-Wunch at **training** time alone (in the first experiment) and at **inference** time alone (in the second experiment). We clarify this below:
> > > > > > > > > 1) In the earlier response (i.e., Drop-DTW: **34.2** vs. Needleman-Wunch: 29.4), we use Needleman-Wunch and Drop-DTW as a differentiable loss function **only** at training time, but we  still use Drop-DTW at inference time in both cases because, as mentioned earlier, the step localization task requires many-to-one matching. Please note that this experimental setup is consistent with the protocol used to generate the results in Table 2 of the main manuscript.
> > > > > > > > > 3) In our latest response (i.e., Drop-DTW: **30.5** vs. Needleman-Wunch: 9.5), we went one step further to make the distinction *crystal* clear. In particular, we use the frozen pre-trained features from [45] and directly compare the performance of Drop-DTW versus Needleman-Wunch as inference procedures to **highlight**, as requested, the essential role of many-to-one matching at inference time that is **only** enabled by Drop-DTW. Please note that this experimental setup is similar to the one used to obtain the results in the first row of Table 1 in the main manuscript.
> > > > > > > > >
> > > > > > > > > In summary, the performance of Drop-DTW in the first experiment (34.2%) is different from that in the second experiment (30.5%), because in the former experiment we actually train the feature embeddings on the CrossTask dataset, while in the second one we use pre-trained features from [45] and use them for inference directly.
> > > > > > > > >
> > > > > > > > > We hope this clarifies the remaining confusion and now casts our contribution as warranting a higher rating.

---

> > > ### Author Response · Authors · 2021-08-31
> > > **Continuing the discussion**
> > >
> > > *Reposting our last comment here for better visibility.*
> > >
> > >
> > > We sincerely thank the reviewer for acknowledging our contribution and positively receiving our response. Indeed, it seems like we are finally converging and we appreciate the opportunity for clarification, which ultimately will make our submission stronger. Here, we address the remaining concern about significance.
> > >
> > >
> > > >* I do want to note that this is a narrower set of claims as compared to what is stated in the submission itself.
> > >
> > > We will integrate our response into the revised manuscript to make the distinctions clearer and more precisely position our work.
> > > ### Significance:
> > > >* importance and significance of the two novel properties mentioned above.
> > >
> > > * **Ability of the proposed method to handle outlier rejection in both sequences**
> > >
> > > Our approach is more general/flexible. It does not require making assumptions that one of the two sequences is outlier-free, which we believe is a **significant** distinguishing aspect between our method and all the alternatives. A concrete application demonstrating the significance of this component is provided in Section 4.4 of our main paper where we show that Drop-DTW can be used in a cross-modal setting with outliers present in both the visual and audio signal (i.e., some things can be heard but not seen, while others can be seen but not heard). In that case, we allow Drop-DTW to drop outliers from both sequences (again using the same optimal, unified framework; see complete algorithm in the supplemental) and achieve superior performance to alternative alignment methods in a completely unsupervised setting.  Notably, we are even competitive with a published fully supervised approach [13].
> > >
> > > To quantitatively demonstrate the importance of Drop-DTW's ability to drop outliers from both sequences, we repeat the experiment from Section 4.4 with the version of Drop-DTW, where the drops are only allowed to one of the sequences, i.e., only audio **or** video, (dubbed Drop-DTW-oneside) and show the results in the table below:
> > >
> > > |    Method    | Drop video| Drop audio | A2V	|V2A	|
> > > |-----------|---|---|-----------|---------|
> > > |DTW	         | no | no | 39.8 | 33.9    	|
> > > |Drop-DTW-oneside| no | yes | 39.0|	33.5|
> > > |Drop-DTW-oneside| yes | no |	38.9|	33.3|
> > > |Drop-DTW (ours) | yes | yes |**41.1**	|**35.8**|
> > >
> > > Clearly, our Drop-DTW outperforms standard DTW and the Drop-DTW-oneside variants. Interestingly, both Drop-DTW-oneside variants perform even worse than standard DTW that cannot drop elements from either sequence. We hypothesize that this follows because standard DTW allows the outliers from both sequences to match to each other, while Drop-DTW-oneside forces all the outliers from one sequence (that is not amenable to drops) to match to only inliers from the other sequence (that admits drops), which creates more erroneous inlier-to-outlier correspondences and results in a more biased training signal. In contrast to both DTW and Drop-DTW-oneside, our Drop-DTW does not have to deal with outlier matching at all, and is able to learn from inlier-to-inlier correspondences only, which is supported by the empirical results in the table.
> > >
> > > * **Ability to do one-to-many matching in the alignment**
> > >
> > > In fact, the ability to do one-to-many matching is **exactly** what enables applications such as step localization. For example, in the case of instructional videos, the video sequence is typically an order of magnitude longer than the step sequence. For example, a typical video from the CrossTask dataset could consist of 200 video clips and contain just five instructional steps. Applying an alignment algorithm with one-to-one matches only (e.g., MVM or Needleman-Wunsch) will match the five instructional steps with only five video clips (i.e., put into one-to-one correspondence), and will be forced to drop the remaining 195 video clips, even if the steps in the video span multiple clips (as typically is the case). Clearly, such a formulation can not be used to find the start and end time of steps in a video, as the start and end time simply collapse into a single point, and most of the video is labeled as "background".  In contrast, Drop-DTW allows to assign multiple video clips to a single step and results in a meaningful temporal segmentation of the video sequence. This means that Drop-DTW does not just achieve better results on the task of step localization, it **enables** to solve this task via alignment. To support these claims, we provide an additional direct comparison to Needleman-Wunsch and Drop-DTW-one-to-one (i.e., Drop-DTW that is restricted to one-to-one matches only) for step localization as an inference procedure in the table below:
> > >
> > >
> > > |           |CrossTask | COIN  | YouCook2 |
> > > |-----------|-----------|---------|-----------|
> > > |	             | IoU    	     | IoU         | IoU    |
> > > |Needleman-Wunch|		9.5|   7.4	 | 11.7	|
> > > |Drop-DTW-one-to-one|		9.7|   8.1	 | 12.0	|
> > > |Drop-DTW	|**30.5**	| **27.7**		|**47.5**|
> > >
> > > We can see that Drop-DTW is indeed significantly better suited for step localization as it improves the Intersection over Union (IoU) metric by three to four times, compared to one-to-one alignment algorithms.
> > >
> > > >* this cannot be determined conclusively from the provided experiment. The two models differ in other aspects as well. In order to prove this out conclusively, authors should disable one-to-many matching in Drop-DTW to see what effect this may have.
> > >
> > > As described and shown above with the Drop-DTW-one-to-one example, turning off the one-to-many matching has a large, negative impact on the instructional video alignment task, as expected, as the very definition of this task requires multiple clips to be matched to one step.  We, therefore, believe that the provided clarification and results speak decisively in favor of our approach.
> > >
> > >
> > >
> > > ### Summary
> > > In summary, our contributions go beyond simple design choices. They actually enable a broad variety of applications, as shown in the paper. With this discussion, we sincerely hope our response convinced the reviewer that our submission meets the axes that they laid out for a NeurIPS paper, and hope this will convince them to raise their rating.
> > > At this point, it appears that we have addressed all the objective concerns; what remains are the subjective remarks raised in the review and responses. If you have further concerns, please share.

---

> ### Author Response · Authors · 2021-09-01
> **Thanks to Reviewer wLu7**
>
> We thank the reviewer for their careful consideration of our responses and for acknowledging: “the important role” inherent in our one-to-many matching approach in the target tasks, the performance difference between extant methods (e.g., Needleman–Wunsch), and the value to the community.
> We sincerely appreciate the increased rating.

---

### Author Response · Authors · 2021-09-01
**Request of Additional Feedback from Reviewer n8aa**

Dear Reviewer [n8aa](https://openreview.net/forum?id=A_Aeb-XLozL&noteId=nIrHxikN-1r),

The concerns about the paper raised in the review were mainly clarifications and have been addressed in our responses.  Corresponding clarifications will be included in the revised manuscript.

We hope the reviewer can provide additional feedback and reconsider their rating.

---

### Author Response · Authors · 2021-09-01
**Discussion Summary (with Reviewer VyHZ)**

We thank Reviewer VyHZ for the very productive dialog, acknowledging our method "has fundamental merits and the new experimental results provided in the discussion are highly valuable and illustrative", the novelty of our method, and "would go along with the acceptance decision". This extensive dialog made our submission stronger, which we believe is the ultimate goal of this review process.

The reviewer's main remaining concern, stopping them from arguing for paper acceptance, is the amount of changes to the final version of the paper.  Here, we highlight the main changes that will be made to the revised manuscript:

- A dedicated subsection in Related Work discussing alignment algorithms that allow to drop outliers or use drop costs, e.g., Needleman-Wunsch and LCSS, and contrasting them with our Drop-DTW. We have included a draft of this revision in our initial common response to all reviewers. We will tighten up our claims of contributions and clarify their significance, e.g., one-to-many matching.
- Inclusion of the experimental comparisons shared in our responses of Drop-DTW to Needleman-Wunsch and LCSS as inference procedures for step localization.

While the changes are important, the list is **concise** and aims at **clarifying the message** already present in the original submission, **rather than adding new contributions**.  Note, all these changes are documented in our responses now without objection, thus leaving no unknowns.  We hope that the reviewer finds the list of modifications compelling and concise and reconsiders their final rating based on the reviewers-AC discussion.

---

### Decision · Program_Chairs · 2021-09-27

**Decision:**

Accept (Poster)

**Comment:**

There has been extensive discussion between the reviewers and authors in the post-rebuttal period to tease apart the novel contributions of this paper and accurately position it wrt prior work.

The authors have committed to repositioning the paper, clarifying the contributions, and thoroughly discussing close prior works in their final version.